# NaVILA: Legged Robot Vision-Language-Action Model for Navigation

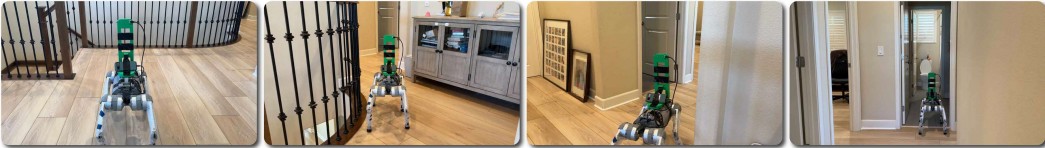

Walk forward, when seeing the stair bars, turn right and walk around the stairs
until reaching the hallway. Turn right and walk along the hallway, stop in front of a bathroom.

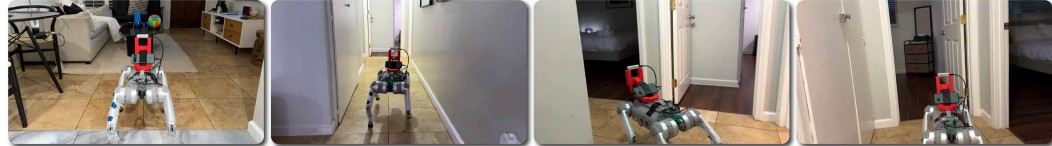

Turn right immediately, walk along the hallway, turn left at the end and enter the most left bedroom.

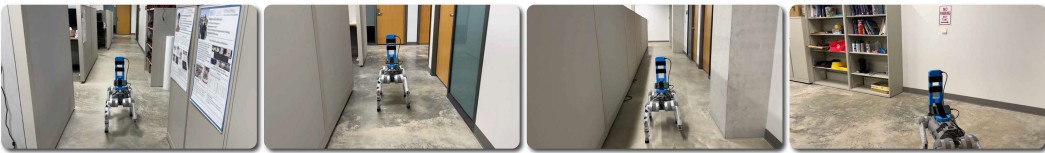

Walk all the way down, turn left at the intersection and find a bookshelf.

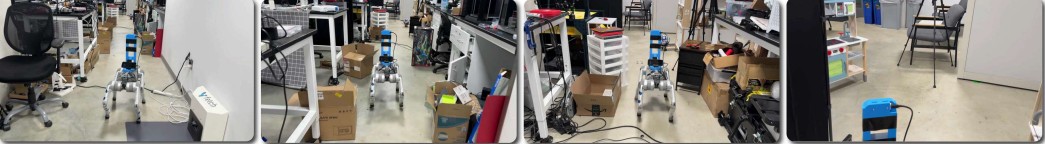

Walk to the other end of the room, turn left and find a toy kitchen set.

Figure 1: Real-world demonstration of NaVILA: Upon receiving human instructions, NaVILA uses a vision-language model to process RGB video frames and employs locomotion skills to execute the task on a robot. The robot successfully handles long-horizon navigation tasks and operates safely in challenging environments.

## ABSTRACT

This paper proposes to solve the problem of Vision-and-Language Navigation with legged robots, which not only provides a flexible way for humans to command but also allows the robot to navigate through more challenging and cluttered scenes. However, it is non-trivial to translate human language instructions all the way to low-level leg joint actions. We propose NaVILA, a 2-level framework that unifies a Vision-Language-Action model (VLA) with locomotion skills. Instead of directly predicting low-level actions from VLA, NaVILA first generates mid-level actions with spatial information in the form of language, (e.g., "moving forward 75cm"), which serves as an input for a visual locomotion RL policy for execution. NaVILA substantially improves previous approaches on existing benchmarks. The same advantages are demonstrated in our newly developed benchmarks with IsaacLab, featuring more realistic scenes, low-level controls, and real-world robot experiments. We show more qualitative results anonymously at https://NaVILA-ICLR.github.io.

# 1 INTRODUCTION

The ability to perform Vision-and-Language Navigation (VLN) has become a foundational component in modern robotics systems. With VLN, a robot is expected to navigate around unseen environments without a provided map following a language instruction (Anderson et al., 2018; Wang et al., 2019; Chaplot et al., 2020a;b;c; Ramrakhya et al., 2022). This not only offers a better interface for humans, but also strengthen cross-scene generalization through languages. In this paper, we further extend the study of VLN with legged robots (e.g., quadruped or humanoid). Using legs instead of wheels allows robots to navigate in more challenging and cluttered scenarios. As the examples shown in Figure 1, our robot can navigate through a crowded office with narrow walkways and scattered desks, or a messy home with toys and other objects on the floor.

To translate language to action, the robot needs to reason about the input language, and perform closed-loop planning as well as low-level control. With the recent advancement in Large Language Models (LLMs) and Vision-Language Models (VLMs), several end-to-end Vision-Language-Action (VLA) systems have been developed (Brohan et al., 2023; Kim et al., 2024; Padalkar et al., 2024). These systems fine-tune a general-propose VLM with large-scale robot manipulation demonstrations to produce low-level actions for control. While unifying reasoning and execution in a single model is fascinating and shows encouraging results, it is worthy to dive deeper into the question: Is there a better way to represent actions beyond the quantized low-level commands? After all, LLMs and VLMs were primarily trained with natural language. Unifying reasoning and execution becomes challenging when we need to convert that reasoning into precise, non-verbal actions.

Inspired by the recent progress on VLM (Chen et al., 2024a; Cheng et al., 2024) for spatial location and distance reasoning, we propose **NaVILA**, a two-level framework for legged robot VLN: A VLM is fine-tuned to output a **mid-level action** (VLA) in the form of language such as "turn right 30 degrees", and a low-level visual locomotion policy is trained to follow this instruction for execution. The mid-level action output of the VLA conveys the location and direction information without the low-level commands. The advantages of this framework are three-fold: (i) By decoupling low-level execution from VLAs, the same VLA can be applied across different robots by swapping the low-level policy; (ii) Representing actions as mid-level language instructions enables training with diverse data sources, including real human videos and reasoning QA tasks. This enhances reasoning capabilities without overfitting outputs to specific low-level commands, and can leverage real-world data for generalization; (iii) NaVILA operates on two distinct timescales: the VLA, typically a large and computationally intensive model, runs at a lower frequency, providing high-level navigation commands; while the locomotion policy operates in real-time. This dual-frequency approach allows the locomotion policy to handle sophisticated obstacle avoidance and increases overall robustness.

To train the VLA, we demonstrate how to (i) Integrate historical context and current observations in VLN within existing VLM frameworks, (ii) Create a specialized navigation prompt tailored for VLN tasks, (iii) Introduce a carefully curated dataset blend designed to enhance VLN generalizability. These strategies allow us to fine-tune a general-purpose image-based VLM into a navigation-focused agent while simultaneously training it on general vision-language datasets, thereby maintaining its broad generalization capabilities.

During training the locomotion skills, we employ a single-stage approach to learn vision-based locomotion policy. We construct a height map from raw LIDAR point clouds and introduce randomization to bridge the sim-to-real gap. This controller takes the output from our VLA model, converts it into command velocities, and tracks these velocities by controlling the positions of the joints. To our knowledge, this is the first end-to-end approach for training visual locomotion skills that are both robust and safe, enabling deployment in real-world, challenging environments (e.g., strong sunlight or near certain transparent surfaces).

In our experiments, we show that our VLA significantly outperforms the state-of-the-arts on classic VLN benchmarks, with over 17% improvement in success rate. To better simulate the challenges of locomotion navigation in VLN, we introduce a new benchmark, VLN-CE-Isaac, using Isaac Sim. This benchmark considers detailed robotic joint movements and interactions with environments, which prior VLN works have not explored. In our VLN-CE-Isaac experiments, our vision-based policy outperforms the blind policy by a significant margin, showing a 14% improvement in success rate. We also demonstrate that our VLA can be deployed across different robots (Unitree Go2 and Unitree H1), each using distinct locomotion skills. Finally, we deploy NaVILA in the real world, exhibiting impressive robustness and achieving an 88% success rate on 25 instructions, including a 75% success rate on complex instructions across diverse scenes.

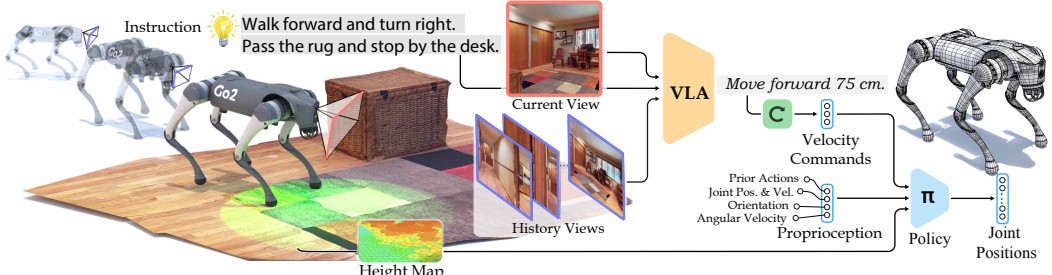

Figure 2: NaVILA is a two-level framework combining high-level visual language understanding with low-level locomotion control. Our VLA model processes single-view images to produce mid-level actions in natural language, which are then converted into precise joint movements by an advanced low-level locomotion policy. This integration allows for strong generalization and adaptability across different real-world environments, and can operate the robot in real-time.

## 2 METHOD

Our VLA model (NaVILA) integrates high-level visual language understanding and action with low-level locomotion control (Figure 2). NaVILA employs a VLM that processes single-view images to generate waypoint instructions in natural language. These instructions are then interpreted by a low-level locomotion policy, which translates them into precise joint movements for real-time robot control. The synergy between the VLM's high-level reasoning and the locomotion policy's execution capabilities enables our method to demonstrate remarkable generalization and adaptability across diverse real-world environments. In the following sections, we detail the components of our approach. We begin by describing how we tame VLMs for high-level VLN in Sec. 2.1, followed by an overview of our robot configuration and low-level locomotion policy in Sec. 2.2.

### 2.1 TAMING VLMS FOR HIGH-LEVEL VISION LANGUAGE NAVIGATION

VLN requires processing video inputs as observations. A common approach to handling video inputs in VLMs is through video encoders. However, recent progress in VLMs has largely been driven by the availability of image-text data. While there have been efforts to extend this success to video encoders, the lack of large, high-quality video-text datasets has limited their pre-training. To address this challenge, we opt for image-based vision-language models in our approach. These models exhibit stronger generalization abilities and possess broader knowledge, making them more suitable for tackling the generalization challenges in VLN. Specifically, we built our approach upon VILA, an image-based VLM pre-trained with interleaved image-text corpus. VILA's pre-training has proven particularly effective for multi-image reasoning, making it especially suitable for VLN tasks where understanding sequential image relationships is critical.

**VILA Preliminary.** VILA consists of three main components: a vision encoder, a projector, and an LLM. The vision encoder processes the input images, converting them into a sequence of visual tokens. These tokens are then downsampled and mapped into the language domain via an MLP projector. Afterward, the projected tokens, along with text tokens, are sent to the LLM for auto-regressive generation. When handling videos, VILA uniformly sampled frames at regular intervals. It puts all the frame information before any text. A typical prompt for describing a video might look like "⟨frame3⟩⟨frame6⟩⟨frame9⟩...Tell me about this video." VILA undergoes a 3-stage training process: first, it pre-trains a connector between the frozen LLM and vision backbones using alignment data (Liu et al., 2023); then it pre-trains both the connector and the LLM using text-image interleaved corpus (Byeon et al., 2022; Zhu et al., 2024); and finally, it fine-tunes all modules (vision encoder, connector, LLM) with instruction tuning data (Liu et al., 2023; 2024).

**Navigation Prompts.** In vision-language navigation tasks, images from different time steps serve two distinct purposes. The image at time step $t$ represents the current observation, which is crucial for a VLN agent to make immediate decisions (e.g., turning right at an intersection or stopping when the goal is reached). On the other hand, frames before time step $t$ are historical frames that function as a memory bank, helping the agent track overall progress (e.g., remembering the starting location, reasoning about places already visited and planning the next step). Uniformly sampling frames at regular intervals, as done in VILA, is not ideal because it doesn't differentiate between these two

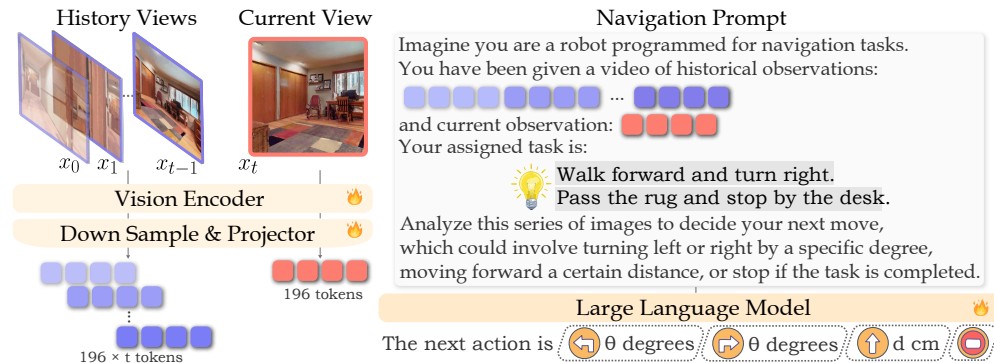

Figure 3: Overview of our VLA framework. We denote the purple blocks (🟣) as memory tokens sampled from historical frames, and the red blocks (🔴) as the current observation tokens. 🔥 denotes trainable parameters.

types of representations. Therefore, we first extract the most recent frame $t$ as the current observation and then uniformly sample frames from the preceding $t-1$ frames, ensuring the first frame is always included. Additionally, since current and historical observations serve different roles, we distinguish them in our task prompt using textual cues like `a video of historical observations:` for memory frames and `current observation:` for the latest frame. Unlike (Zhang et al., 2024), we avoid introducing additional special tokens that could complicate the LLM's learning process. Instead, we adhere to our design principle of keeping both the input and output of LLM in the language domain to fully leverage the reasoning capabilities of the pre-trained LLM. By integrating these tokens for historical and current observations with the navigation instruction, we construct a navigation task prompt, as shown in Figure 2.

**Supervised Fine-tuning Data Blend.** Effective Supervised Fine-tuning (SFT) data is crucial for developing a robust vision-language action model. Such a model should be specialized for an embodied task yet avoid overfitting to specific actions. It should also generalize well to real-world scenarios while retaining broad-world knowledge. Thanks to NaVILA's modular framework design, which offers exceptional scalability and adaptability, it is straightforward to integrate new data sources into our pipeline. This flexibility allows us to consider diverse data sources to improve generalizability for navigation. We designed our SFT data blend from four perspectives: (1) Navigational data from simulations, (2) Navigational data from real videos, (3) Auxiliary navigational data, and (4) General VQA datasets.

First, we focus on navigational data in simulations. Currently, there are limited options for VLN datasets in continuous environments, with only R2R-CE (Krantz et al., 2020b) and RxR-CE (Ku et al., 2020) available. These datasets provide sparse path points converted from discrete VLN versions. We utilize both datasets within the Habitat simulator, employing a shortest path follower to generate action sequences that adhere to the geodesic shortest path. This results in step-wise navigation videos, where each sample in our dataset comprises a $t+1$ frames video and the corresponding oracle action at time step $t$. To encourage the LLM to generate continuous value labels for distances and angles, we merge consecutive actions (e.g., combining two forward 25 cm steps into a single forward 50 cm step), with a maximum of three consecutive actions. This merging process has two key advantages: it reduces the dataset size for more efficient processing, and it helps prevent overfitting by introducing greater diversity in the actions. Additionally, to address label imbalance, particularly the underrepresentation of the stop action, we apply a rebalancing technique for a more even distribution. For all navigation-specific data, we apply the previously described frame extraction strategy and navigation task prompt.

Second, we incorporate navigational data from real videos. Specifically, we collect 2K egocentric human touring videos from YouTube, using these as a rich source of real data for learning robot navigation from human behavior. The videos are first processed into 20K trajectories through entropy-based sampling (Lin et al., 2023) to ensure representative and diverse samples. Then we estimate camera poses with Mast3R (Leroy et al., 2024) to extract step-wise actions and generate natural language instructions for each trajectory through VLM captioning and LLM rephrasing.

Third, to improve scene understanding and address the limited instructions in current R2R-CE and RxR-CE, we incorporate auxiliary navigational datasets. Following (Zhang et al., 2024), we use

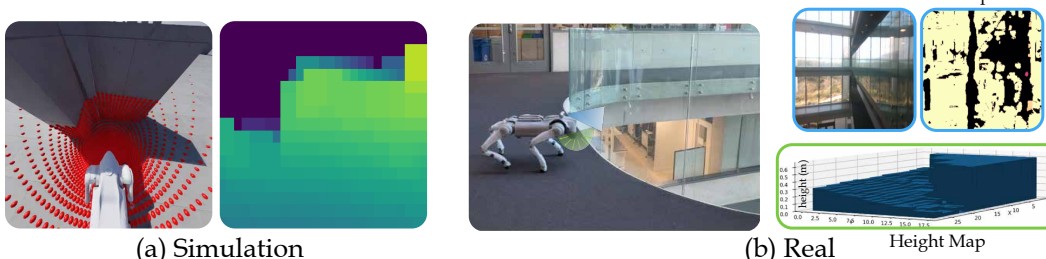

(a) Simulation  (b) Real

Figure 4: Height map reconstruction from the point cloud. (**a**) Go2 robot tracks linear and angular velocity commands while avoiding collision with obstacles in simulation. The red dots represent the LiDAR point cloud, raycasting from the sensor's center towards the terrain mesh. The left image is a preprocessed height map from the LiDAR data, with values clipped according to real-world sensor constraints. Darker colors represent lower values. (**b**) Safe robot locomotion near a transparent glass surface. The top-down height map clearly detects the glass, while the forward-facing depth and RGB images struggle to capture it.

augmented instructions from EnvDrop (Tan et al., 2019) and introduce an auxiliary task of navigation trajectory summarization. Specifically, given a trajectory video, we sample frames by retaining the first frame and uniformly sampling the remaining ones as historical frames, and use the annotated instructions as labels. The LLM is then tasked with describing the robot's navigation trajectory based on these frames. To further encourage spatial scene understanding, we integrate the ScanQA (Azuma et al., 2022) dataset, which features real-world 3D scan QA pairs involving human-edited questions and free-form answers grounded to 3D objects within each scene. For training, we utilize the multi-view RGB images from the raw scans to support this task.

Finally, to maintain the model's general capabilities, we include general video/image VQA datasets from (Liu et al., 2024; Chen et al., 2024d; Maaz et al., 2024). This comprehensive dataset design enables NaVILA to generalize effectively to novel scenes and real-world environments.

**Training and Inference Paradigm.** Our training process begins with the stage two model of VILA, which has already undergone visual language corpus pre-training. We then apply our SFT data blend to train the entire VLM for one epoch, following standard practices. During this training, all three components—vision encoder, connector, and LLM—are unfrozen. For the inference phase, we implement a regular expression parser (Kearns, 1991), to extract action types (such as forward or turn left) and their corresponding arguments (like specific distance or angles) from the LLM output. This method has demonstrated effectiveness in both simulated environments and real-world experiments, where we empirically found that all actions throughout all experiments are successfully matched and mapped.

## 2.2 Visual Locomotion Policy

In this section, we begin with a brief overview of the Go2 robot dog, the experimental platform used in this work. Next, we describe the development of the end-to-end vision-based control policy, which interprets high-level language navigation commands from the VLM and converts them into precise joint movements. This control policy is trained in the Isaac Sim simulator using Isaac Lab (Mittal et al., 2023) and then directly deployed to the real-world robot.

**Go2 Robot.** As shown in Figure 4, the robot is equipped with a LiDAR sensor mounted at the base of its head, broadcasting point clouds at a frequency of 15Hz. The robot features 18 degrees of freedom (DoFs), comprising 6 DoFs for its base and 3 DoFs for each of its four legs. In the policy training process, we left the 6 DoFs on the base unconstrained so that the policy only controls the 12 joint motors on the legs.

**Interpreting High-level Commands** As in our formulation, VLM outputs a fixed set of actionable words, such as {move forward, turn left, turn right, stop}, we casts these instructions to fixed command velocities $\{0.5\,\mathrm{m\,s^{-1}}, \frac{\pi}{6}\,\mathrm{rad\,s^{-1}}, -\frac{\pi}{6}\,\mathrm{rad\,s^{-1}}, 0\}$ and execute with corresponding time durations to align with the specific VLM value.

**Low-level Action and Observation Space.** The action space $\mathbf{a}$ of the control policy is defined as the desired joint position $q^d \in \mathbb{R}^{12}$, which are converted into torque input for the simulator using the stiffness and dampness. We adopt PPO algorithm (Schulman et al., 2017) to train the policy. During

Table 1: Comparison with state-of-the-art methods on the Val-Unseen split of R2R-CE (Krantz et al., 2020b) and RxR-CE (Ku et al., 2020). * indicates methods using the waypoint predictor from Hong et al. (2022). NaVILA achieves remarkable performance among methods using only single-view RGB input and shows competitive results compared to those using panorama, depth, or odometry sensors.

| | Observation | | | | R2R Val-Unseen | | | | RxR Val-Unseen | | | |
| --- | --- | --- | --- | --- | --- | --- | --- | --- | --- | --- | --- | --- |
| | S.RGB | Pano. | Depth | Odo. | NE ↓ | OS ↑ | SR ↑ | SPL ↑ | NE ↓ | SR ↑ | SPL ↑ | nDTW ↑ |
| HPN+DN* (Krantz et al., 2021) | | ✓ | ✓ | ✓ | 6.31 | 40.0 | 36.0 | 34.0 | - | - | - | - |
| CMA* (Hong et al., 2022) | | ✓ | ✓ | ✓ | 6.20 | 52.0 | 41.0 | 36.0 | 8.76 | 26.5 | 22.1 | 47.0 |
| VLN○BERT* (Hong et al., 2022) | | ✓ | ✓ | ✓ | 5.74 | 53.0 | 44.0 | 39.0 | 8.98 | 27.0 | 22.6 | 46.7 |
| Sim2Sim* (Krantz & Lee, 2022) | | ✓ | ✓ | ✓ | 6.07 | 52.0 | 43.0 | 36.0 | - | - | - | - |
| GridMM* (Wang et al., 2023c) | | ✓ | ✓ | ✓ | 5.11 | 61.0 | 49.0 | 41.0 | - | - | - | - |
| Ego²-Map* (Hong et al., 2023a) | | ✓ | ✓ | ✓ | 5.54 | 56.0 | 47.0 | 41.0 | - | - | - | - |
| DreamWalker* (Wang et al., 2023a) | | ✓ | ✓ | ✓ | 5.53 | 59.0 | 49.0 | 44.0 | - | - | - | - |
| Reborn* (An et al., 2022) | | ✓ | ✓ | ✓ | 5.40 | 57.0 | 50.0 | 46.0 | 5.98 | 48.6 | 42.0 | 63.3 |
| ETPNav* (An et al., 2024) | | ✓ | ✓ | ✓ | 4.71 | 65.0 | 57.0 | 49.0 | 5.64 | 54.7 | 44.8 | 61.9 |
| HNR* (Wang et al., 2024) | | ✓ | ✓ | ✓ | 4.42 | 67.0 | 61.0 | 51.0 | 5.50 | 56.3 | 46.7 | 63.5 |
| BEVBert* (An et al., 2023) | | ✓ | ✓ | ✓ | 4.57 | 67.0 | 59.0 | 50.0 | 4.00 | 68.5 | - | 69.6 |
| HAMT+ScaleVLN* (Wang et al., 2023d) | | ✓ | ✓ | ✓ | 4.80 | - | 55.0 | 51.0 | - | - | - | - |
| AG-CMTP (Chen et al., 2021a) | | ✓ | ✓ | ✓ | 7.90 | 39.0 | 23.0 | 19.0 | - | - | - | - |
| R2R-CMTP (Chen et al., 2021a) | | ✓ | ✓ | ✓ | 7.90 | 38.0 | 26.0 | 22.0 | - | - | - | - |
| LAW (Raychaudhuri et al., 2021) | ✓ | | ✓ | ✓ | 6.83 | 44.0 | 35.0 | 31.0 | 10.90 | 8.0 | 8.0 | 38.0 |
| CM2 (Georgakis et al., 2022) | ✓ | | ✓ | ✓ | 7.02 | 41.0 | 34.0 | 27.0 | - | - | - | - |
| WS-MGMap (Chen et al., 2022) | ✓ | | ✓ | ✓ | 6.28 | 47.0 | 38.0 | 34.0 | - | - | - | - |
| AO-Planner (Chen et al., 2024b) | | ✓ | ✓ | | 5.55 | 59.0 | 47.0 | 33.0 | 7.06 | 43.3 | 30.5 | 50.1 |
| Seq2Seq (Krantz et al., 2020a) | ✓ | | ✓ | | 7.77 | 37.0 | 25.0 | 22.0 | 12.10 | 13.9 | 11.9 | 30.8 |
| CMA (Krantz et al., 2020a) | ✓ | | ✓ | | 7.37 | 40.0 | 32.0 | 30.0 | - | - | - | - |
| RGB-Seq2Seq (Krantz et al., 2020a) | ✓ | | | | 10.10 | 8.0 | 0.0 | 0.0 | - | - | - | - |
| RGB-CMA (Krantz et al., 2020a) | ✓ | | | | 9.55 | 10.0 | 5.0 | 4.0 | - | - | - | - |
| NaVid (Zhang et al., 2024) | ✓ | | | | 5.47 | 49.0 | 37.0 | 35.0 | - | - | - | - |
| NaVILA | ✓ | | | | 5.22 | 62.5 | 54.0 | 49.0 | 6.77 | 49.3 | 44.0 | 58.8 |

training, the critic observes the privileged environment and generates a value function to update the actor. The actor then only receives sensor data available in the real world. The observation space of the critic $\mathbf{o}^c$ contains the proprioception and velocity command at the current time step $t$ and a privileged terrain height scan around the robot. The proprioceptive data includes robot linear and angular velocity, orientation, joint positions, joint velocities, and the previous action. In the actor's observation space $\mathbf{o}^a$, linear velocity is excluded, as it is unavailable in the real world, and instead, a history of proprioceptive data is used to infer this information implicitly. The robot perceives the surrounding terrain using a heightmap from the LiDAR sensor.

**Incorporating Height Map from LiDAR Point Cloud.** Given LiDAR's superior ability to detect transparent objects and robust performance under strong sunlight, we chose the manufacturer-provided LiDAR as the primary sensor for perceiving the robot's surroundings and ensuring safe navigation. The Unitree L1 generates point clouds with a wide field of view of $360° \times 90°$, from which we create a 2.5D height map based on the parameters listed in Table 13. For each voxel grid, the lowest value within the range is selected, and a maximum filter is then applied over the last 5 lidar point clouds to smooth the resulting height map.

**Training.** Different from most existing works (Lee et al., 2020; Miki et al., 2022; Margolis et al., 2022; Lee et al., 2024) that utilize the two-stage teacher-student training paradigm, we adopt a single-stage manner to train the locomotion policy. Compared to two-stage training, single-stage RL is more time-efficient as it eliminates the need for policy distillation. Additionally, the policy interacts directly with the environment, allowing it to explore and potentially discover novel strategies. With the support of ray-casting in Isaac Lab, our vision-based RL policy training achieves a high throughput over 60K FPS on an RTX 4090 GPU. Training details are included in Appx. F.

## 3 EXPERIMENTS

We conduct experiments to answer the following questions: (1) How does our VLA's performance compare to state-of-the-art methods in VLN-CE benchmarks and general spatial scene understanding tasks? (Section. 3.1) (2) How to evaluate locomotion navigation in simulators, and how effective and flexible is NaVILA in these scenarios? (Section. 3.2) (3) Can NaVILA pipeline be successfully deployed in real robot VLN experiments? (Section. 3.3)

Table 2: Cross-dataset performance on the RxR-CE (Ku et al., 2020) Val-Unseen split. All results are obtained without training on the RxR-CE training set. NaVILA significantly outperforms NaVid (Zhang et al., 2024), the current single-view state-of-the-art.

| | Observation | | | RxR Val-Unseen | | | |
|---|---|---|---|---|---|---|---|
| | S.RGB | Depth | Odo. | NE ↓ | OS ↑ | SR ↑ | SPL ↑ |
| LAW (Raychaudhuri et al., 2021) | ✓ | ✓ | ✓ | 10.87 | 21.0 | 8.0 | 8.0 |
| CM2 (Georgakis et al., 2022) | ✓ | ✓ | ✓ | 8.98 | 25.3 | 14.4 | 9.2 |
| WS-MGMap (Chen et al., 2022) | ✓ | ✓ | ✓ | 9.83 | 29.8 | 15.0 | 12.1 |
| Seq2Seq (Krantz et al., 2020a) | ✓ | ✓ | | 11.8 | 5.02 | 3.51 | 3.43 |
| CMA (Krantz et al., 2020a) | ✓ | ✓ | | 11.7 | 10.7 | 4.41 | 2.47 |
| RGB-Seq2Seq (Zhang et al., 2024) | ✓ | | | 11.2 | 12.2 | 0.0 | 0.0 |
| RGB-CMA (Zhang et al., 2024) | ✓ | | | 9.55 | 14.8 | 0.0 | 0.0 |
| $A^2$NAV (Chen et al., 2023) | ✓ | | | - | - | 16.8 | 6.3 |
| NaVid (Zhang et al., 2024) | ✓ | | | **8.41** | 34.5 | 23.8 | 21.2 |
| NaVILA | ✓ | | | 8.78 | **46.8** | **34.3** | **28.2** |

Table 3: Evaluation of spatial scene understanding performance on the ScanQA dataset (Azuma et al., 2022) Validation split. NaVILA outperforms current state-of-the-art VLA models and demonstrates comparable or superior performance to other 3D LMMs (LMMs) that require additional input, such as depth or camera pose. Note that ∗ indicates 3D LMMs that require task-specific fine-tuning on the ScanQA dataset.

| | ScanQA Validation | | | | |
|---|---|---|---|---|---|
| | Bleu-4 ↑ | Rouge ↑ | Cider ↑ | Meteor ↑ | EM ↑ |
| *Task-specific Specialist* | | | | | |
| VoteNet+MCAN (Yu et al., 2019) | 6.2 | 29.8 | 54.7 | 11.4 | 17.3 |
| ScanRefer+MCAN (Yu et al., 2019) | 7.9 | 30.0 | 55.4 | 11.5 | 18.6 |
| ScanQA (Azuma et al., 2022) | 10.1 | 33.3 | 64.9 | 13.1 | 21.0 |
| 3D-VisTA (Zhu et al., 2023) | 10.4 | 35.7 | 69.6 | 13.9 | 22.4 |
| *3D Large Multi-modal Models* | | | | | |
| 3D-LLM$_{(flamingo)}$∗ (Hong et al., 2023b) | 7.2 | 32.3 | 59.2 | 12.2 | 20.4 |
| 3D-LLM$_{(BLIP2-flant5)}$∗ (Hong et al., 2023b) | 12.0 | 35.7 | 69.4 | 14.5 | 20.5 |
| LL3DA∗ (Chen et al., 2024e) | 13.5 | 37.3 | 76.8 | 15.9 | - |
| Chat-3Dv2∗ (Huang et al., 2024a) | 14.0 | - | 87.6 | - | - |
| Scene-LLM∗ (Fu et al., 2024) | 12.0 | 40.0 | 80.0 | 16.6 | 27.2 |
| LEO (Huang et al., 2024b) | 13.2 | 49.2 | 101.4 | 20.0 | 24.5 |
| *2D Vision-Langauge-Action Models* | | | | | |
| NaviLLM (Zheng et al., 2024) | 12.0 | 38.4 | 75.9 | 15.4 | 23.0 |
| NaVILA (8 frames) | 14.8 | 46.4 | 95.1 | 18.7 | 27.0 |
| NaVILA (64 frames) | **16.9** | **49.3** | **102.7** | **20.1** | **28.6** |

## 3.1 VLM Performance

**VLN-CE Benchmarks.** We evaluate our VLM on the VLN-CE benchmarks, which provide continuous environments for executing navigational actions in reconstructed photorealistic indoor scenes. We focus on the val-unseen split in both R2R (Room-to-Room) and RxR (Room-across-Room) datasets within VLN-CE, as these are the two most recognized benchmarks in VLN. We employ the following widely used evaluation metrics for VLN tasks: Navigation Error (NE), Oracle Success Rate (OS), Success Rate (SR), Success-weighted Path Length (SPL), and normalize dynamic time wrapping (nDTW). We show results in Table 1, where NaVILA significantly surpasses all baselines under identical conditions (i.e., single-view RGB) in both benchmarks using a single model. Notably, this also marks the first time a VLN agent, trained solely on single-view RGB input, achieves comparable or superior results to models that use panoramic views, odometry, or simulator-pretrained waypoint predictors. This suggests that NaVILA's strong generalization capabilities can effectively compensate for the limited observations in RGB views or odometry.

To evaluate the cross-dataset performance, we follow (Zhang et al., 2024) by training NaVILA exclusively on R2R samples, while leaving out the RxR training set. We then evaluate its zero-shot performance on the RxR Val-Unseen split. As shown in Table 2, our method significantly outperforms NaVid, the current state-of-the-art model, with a substantial 10% improvement in SR.

Table 4: VLN-CE-Isaac evaluation.

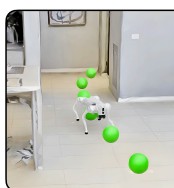

Figure 5: VLN-CE-Isaac Benchmark

| | Low-level Observation | | | VLN-CE-Isaac | | | |
|---|---|---|---|---|---|---|---|
| | Proprio. | LiDAR | Height Scan | NE ↓ | OS ↑ | SR ↑ | SPL ↑ |
| Oracle | | | | 5.25 | 59.8 | 51.3 | 46.9 |
| NaVILA-Go2-Blind | ✓ | | | 6.03 | 49.0 | 36.2 | 33.3 |
| NaVILA-Go2-Vision | ✓ | | ✓ | **5.49** | **58.7** | **50.2** | **45.5** |
| NaVILA-H1-Blind | ✓ | | | 7.67 | 33.3 | 24.4 | 21.0 |
| NaVILA-H1-Vision | ✓ | | ✓ | **5.86** | **54.6** | **45.3** | **40.3** |

Table 5: Real-world experiments conducted in three environments (`Laboratory`, `House`, and `Outdoor`). Simple and Complex refer to simple and complex instruction-following tasks, respectively.

| | Laboratory | | | | House | | | | Outdoor | | | |
|---|---|---|---|---|---|---|---|---|---|---|---|---|
| | Simple | | Complex | | Simple | | Complex | | Simple | | Complex | |
| | NE↓ | SR↑ | NE↓ | SR↑ | NE↓ | SR↑ | NE↓ | SR↑ | NE↓ | SR↑ | NE↓ | SR↑ |
| GPT-4o (OpenAI, 2024) | 2.01 | 0.67 | 2.38 | 0.33 | 1.49 | 0.53 | 3.00 | 0.00 | - | 0.67 | - | 0.50 |
| NaVILA | **1.29** | **1.00** | **1.76** | **0.80** | **1.15** | **1.00** | **1.76** | **0.67** | - | **1.00** | - | **0.83** |

**Spatial Scene Understanding Benchmarks.** As a general navigation agent, robust spatial scene understanding (e.g., object localization, referring, and spatial reasoning) is crucial. To evaluate NaVILA's capabilities in scene understanding, we conduct evaluations on the ScanQA Validation benchmark, a widely used dataset for 3D Question Answering. ScanQA is based on real-world scans, and we use multi-view images from these scans as input to query NaVILA for answers. As shown in Table 3, NaVILA significantly outperforms the previous state-of-the-art model, NaviLLM (Zheng et al., 2024), by a substantial margin (20 points higher on the CIDEr score). Moreover, when using 64 frames, NaVILA's performance demonstrates superior performance compared to state-of-the-art 3D-based large multi-modal models (Huang et al., 2024b; Fu et al., 2024). This is particularly noteworthy as these other models require either 3D scans or RGBD data with camera poses as inputs, while our method achieves better results with less observation.

## 3.2 LEGGED ROBOT NAVIGATION PERFORMANCE IN SIMULATION

**High-fidelity VLN-CE-Isaac Benchmark.** Currently, there are no VLN-CE benchmarks tailored specifically for legged robots. Existing benchmarks (Krantz et al., 2020b; Ku et al., 2020) for vision-language navigation rely on the Habitat (Savva et al., 2019) simulator, which focuses on high-level planning without addressing precise low-level robotic control. For instance, agents in Habitat can navigate through narrow gaps, such as a 10 cm space between two sofas, which is impractical for legged robots like quadrupeds or humanoids. To overcome this limitation, we introduce a new benchmark, VLN-CE-Isaac, built on Isaac Sim. Isaac Sim's high-fidelity simulation captures detailed robotic joint movements and interactions with the environment, enabling comprehensive evaluations of the entire navigation pipeline, from high-level planning to precise robotic execution. We incorporate the same scenes from R2R, with robots deployed in the environment, as shown in Figure 5. From the 1,839 trajectories in the R2R Val-Unseen split, we select 1,077 traversable trajectories with high-quality meshes to ensure realistic navigation scenarios. For consistency, we evaluate performance using the same metrics as prior work.

Notably, VLN-CE-Isaac is compatible with a variety of robotic platforms. To demonstrate this flexibility, we test our NaVILA model on a Unitree Go2 robot and also a Unitree H1 robot within the benchmark. To highlight the effectiveness of the vision-based policy, we compare it against a proprioception-only (*blind*) policy. As shown in Table 4, the vision-based policy outperforms the blind policy by 14% in Success Rate in Go2 settings and 21% in H1 settings, owing to its superior obstacle avoidance capability. We also compare NaVILAs with a baseline using Oracle's low-level policy (assuming perfect command execution without realistic physics). Results show a 15% lower success rate on the Go2 setup and a 27% lower success rate on the H1 setup when Oracle policy is not presented. These performance gaps highlight the increased challenges and realism introduced by our benchmark. Additionally, we also observe that the success rate of NaVILA on the H1 robot is significantly lower than on the Go2, which is expected due to the larger size of the humanoid robot.

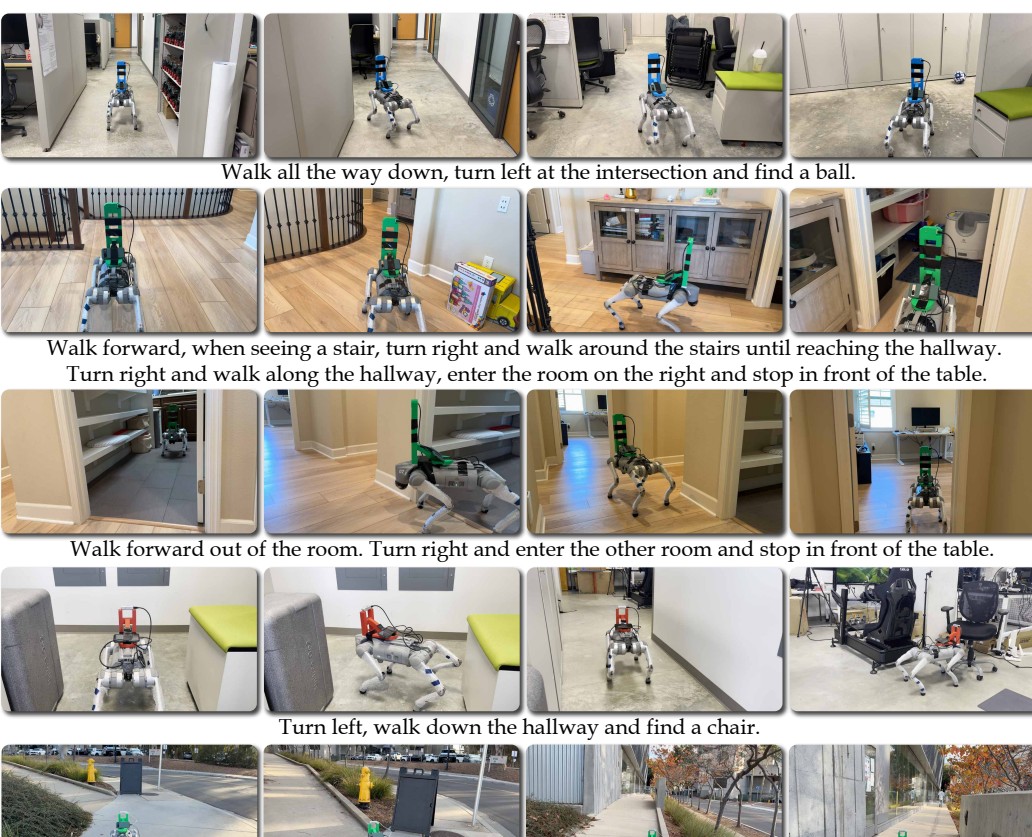

Walk all the way down, turn left at the intersection and find a ball.

Walk forward, when seeing a stair, turn right and walk around the stairs until reaching the hallway.
Turn right and walk along the hallway, enter the room on the right and stop in front of the table.

Walk forward out of the room. Turn right and enter the other room and stop in front of the table.

Turn left, walk down the hallway and find a chair.

Move forward along the way. Turn left at the yellow fire hydrant.
Go forward along the slope and stop in front of the door.

Figure 6: Qualitative results from the real-world deployment of NaVILA. The robot demonstrates the ability to solve long-horizon navigation tasks, following human instructions across various real-world environments.

### 3.3 REAL WORLD EVALUATION

We then conduct experiments in the real world, in which we use a total of 25 instructions, covering both simple and complex tasks, across three types of environments: laboratory space, house, and outdoor open environments. Simple instructions consisted of one or two navigation commands, where the robot did not need to navigate between rooms (e.g., 'Go to the chair and stop'). In contrast, complex instructions involved three or four commands, requiring the robot to traverse multiple rooms or landmarks (e.g., 'Walk out of the room, turn right, enter the room in front of you, and stop at the table'). We use standard metrics (SR and NE) and compare NaVILA against GPT-4o, a state-of-the-art VLM known for its strong generalizability. As shown in Table 5, NaVILA significantly outperforms GPT-4o in both SR and NE. Our qualitative results are presented in Figure 1 and Figure 6. These results highlight the effectiveness of NaVILA in bridging the gap between vision-language understanding and real-world navigation tasks.

## 4 RELATED WORK

**Vision-Language Navigation.** Visual navigation has been a long-standing research topic in robotics for decades (Moravec, 1980; Elfes, 1987; Thrun et al., 2001; Gervet et al., 2023). Classical approaches often rely on pre-computed maps (Thrun et al., 1999) or build geometric maps of the environment using depth sensors (Newcombe et al., 2011) or monocular RGB cameras while localizing the robot simultaneously (SLAM) (Davison et al., 2007; Jones & Soatto, 2011). In recent years, learning-based approaches with Imitation Learning (Chaplot et al., 2018; Codevilla et al., 2018) and Reinforcement Learning (Mnih et al., 2015; Lillicrap, 2015) have not only shown impressive results but also enabled wider applications including vision-and-language navigation.

Vision-Language Navigation (VLN) is a fundamental challenge in embodied AI, where agents navigate complex environments using visual cues and natural language instructions. The field has evolved significantly over time. Early research (Anderson et al., 2018; Ku et al., 2020; Qi et al., 2020) focused on discrete navigation in simulated environments like MP3D (Chang et al., 2017), where agents teleport between predefined nodes on a navigation graph (Fried et al., 2018; Ma et al., 2019; Tan et al., 2019; Ke et al., 2019; Hong et al., 2020; Chen et al., 2021b; 2024c; Zhou et al., 2024). As foundation models advanced, many VLN systems improved dramatically by leveraging large-scale pre-trained models (Li et al., 2019; Majumdar et al., 2020) and pre-training techniques (Guhur et al., 2021; Wang et al., 2023d; Kamath et al., 2023), approaching human-level performance in this setting. However, this setup emphasized high-level decision-making while neglecting the challenges of underlying motion control. Recently, research (Raychaudhuri et al., 2021; Chen et al., 2022; Georgakis et al., 2022; Chen et al., 2024b; Zhang et al., 2024) has shifted towards continuous environments (VLN-CE (Krantz et al., 2020a)) using simulators like Habitat (Savva et al., 2019). This introduces greater complexity, as agents must perform mid-level actions such as moving forward or rotating, rather than teleporting between nodes. To bridge the gap between discrete and continuous navigation, some approaches (Irshad et al., 2021; Krantz & Lee, 2022; An et al., 2023; 2024) use simulator pre-trained waypoint models (Hong et al., 2022; Krantz et al., 2021) that predict candidate positions around the agent and have shown significant performance gains. However, they often struggle to generalize due to their reliance on simulator-specific data. Additionally, the candidate positions predicted by these models only cover nearby locations and do not account for low-level motion planning or obstacle avoidance. In this paper, we aim to advance VLN towards real-world robotics applications, particularly for challenging legged robots. We propose a model that handles both high-level decision-making and generates low-level actions to control the robot's full motion. Additionally, we introduce a new VLN benchmark built on Isaac Sim, offering a more realistic simulation environment, which we believe will benefit future work in VLN.

**Robot Foundation Models.** Robot foundation models aim to provide a unified framework that processes inputs from various modalities, such as vision and language, and directly outputs actions to enable robots to perform complex tasks. Existing works (Brohan et al., 2023; Team et al., 2024; Kim et al., 2024) trained on large-scale robotic dataset to get general robot policies, but mainly focusing on manipulation tasks. Doshi et al. (2024) and Yang et al. (2024) proposed end-to-end visual-language cross-embodiment models for different robotic tasks. As for legged robots, Ding et al. (2024) proposed a unified model to leverage vision and language inputs and generate executable low-level actions. However, these methods struggle to understand complex instructions which are crucial for navigation tasks. Based on this, we propose a VLA model specifically designed for navigation tasks. Our model generates high-level action commands, which are then executed by a low-level policy. This approach enables the robot to interpret complex instructions and navigate effectively towards the goals.

**Legged Robot Locomotion Learning.** Legged robot learning for locomotion navigation focuses on enabling robots to traverse various terrains. Previous works (Wang et al., 2023b; Long et al., 2024) rely solely on robot's proprioceptive information struggle in scenarios like obstacle avoidance. Other end-to-end vision-based approaches (Kareer et al., 2023; Yang et al., 2022; Imai et al., 2022; Yang et al., 2023) are vulnerable to extreme environmental conditions, such as intense sunlight, due to the limitations of sensors. Lee et al. (2020) and Miki et al. (2022) incorporate LiDAR sensors in addition to depth cameras to improve terrain sensing, but rely on time-inefficient two-state training. To overcome these limitations, we propose a single-stage RL framework that integrates LiDAR sensing inputs, allowing the robot to directly learn from interacting with the environments for more efficient learning and robust performance in complex scenarios.

## 5 CONCLUSION

We introduce NaVILA, a powerful two-level framework that unifies vision-language models (VLMs) with locomotion skills for generic navigation tasks. NaVILA generates high-level, language-based commands, while a real-time locomotion policy handles obstacle avoidance. This dual-frequency design improves robustness and flexibility across different robots. By preserving reasoning capabilities through language-based actions, NaVILA avoids overfitting and can be trained on broader tasks. In experiments, NaVILA shows a 17% improvement on classic VLN benchmarks, outperforms vision-blind policies in our new VLN-CE-Isaac1K benchmark, and demonstrates strong real-world performance across diverse scenes. The source code will be released upon publication.

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

APPENDIX TABLE OF CONTENTS

# A ABLATION STUDY ON DIFFERENT SIMULATION DATA BLENDS IN VLA TRAINING

We perform an ablation study to assess the impact of different simulation data blends on training VLA. As shown in Table 6, training navigation data without label rebalancing leads to a significant drop in performance. Additionally, training VLA exclusively on RxR data demonstrates reasonable cross-dataset performance on R2R-CE, supporting our observations in Table 2. Lastly, we investigate whether excluding RxR dataset degrades R2R-CE performance. The results suggest that the RxR dataset does not significantly contribute to the R2R-CE performance.

Table 6: Results on R2R-CE using different data blends.

| | R2R-CE Val Unseen | | | |
| --- | --- | --- | --- | --- |
| | NE $\downarrow$ | OSR $\uparrow$ | SR $\uparrow$ | SPL $\uparrow$ |
| NaVILA (w/o Label Balancing) | 7.82 | 47.5 | 30.0 | 25.1 |
| NaVILA (w/ RxR only) | 7.57 | 40.8 | 31.5 | 27.8 |
| NaVILA (w/o RxR) | 6.11 | 57.0 | 47.7 | 42.4 |
| NaVILA $_{Baseline}$ | **5.37** | **57.6** | **49.7** | **45.5** |

# B ABLATION STUDY ON REAL VIDEO DATA IN VLA TRAINING

We conduct ablation studies in both simulation and real-world settings to evaluate the impact of incorporating real-world data from YouTube human touring videos. As shown in Table 7, the simulation results demonstrate significant performance improvements, with approximately 5% gains in OS, SR, and SPL metrics. Similarly, the real-world experiments, detailed in Table 8, show increased success rates and reduced navigation errors. These findings highlight the scalability and effectiveness of NaVILA's framework, which enables straightforward data integration from different sources.

Table 7: Results on R2R-CE using additional data from human touring videos.

| | R2R-CE Val Unseen | | | |
| --- | --- | --- | --- | --- |
| | NE $\downarrow$ | OSR $\uparrow$ | SR $\uparrow$ | SPL $\uparrow$ |
| NaVILA (w/o Real Videos) | 5.37 | 57.6 | 49.7 | 45.5 |
| NaVILA | **5.22** | **62.5** | **54.0** | **49.0** |

Table 8: Ablation study on real video data in real-world experiments. Simple and Complex refer to simple and complex instruction-following tasks, respectively.

| | Laboratory | | | | House | | | | Outdoor | | | |
| --- | --- | --- | --- | --- | --- | --- | --- | --- | --- | --- | --- | --- |
| | Simple | | Complex | | Simple | | Complex | | Simple | | Complex | |
| | NE $\downarrow$ | SR $\uparrow$ | NE $\downarrow$ | SR $\uparrow$ | NE $\downarrow$ | SR $\uparrow$ | NE $\downarrow$ | SR $\uparrow$ | NE $\downarrow$ | SR $\uparrow$ | NE $\downarrow$ | SR $\uparrow$ |
| NaVILA (w/o Real Videos) | 2.00 | 0.60 | 1.81 | 0.73 | 2.17 | 0.47 | 2.32 | 0.40 | - | 0.00 | - | 0.00 |
| NaVILA | **1.29** | **1.00** | **1.76** | **0.80** | **1.15** | **1.00** | **1.76** | **0.67** | **-** | **1.00** | **-** | **0.83** |

# C ABLATION STUDY ON DIFFERENT MEMORY SIZE

We conduct an ablation study to evaluate the impact of memory size (number of history frames) on two tasks: the navigation task using R2R-CE and the spatial understanding task using ScanQA. The results in Table 9 show that for R2R-CE, 8 frames are sufficient to cover most instruction horizons, with limited performance gains from increasing the memory size. In contrast, ScanQA requires a finer-grained memory to recall details such as spatial information of previously seen objects. The results in Table 10 show that NaVILA's performance consistently improves with a larger memory size. Notably, with a memory size of 64 frames, NaVILA outperforms state-of-the-art 3D-LLMs (Scene-LLM (Fu et al., 2024) and LEO (Huang et al., 2024b)) across all metrics. For real-world experiments, we use an 8-frame memory size due to latency constraints. While larger memory sizes could potentially improve performance, we leave it as future work.

Table 9: Ablation study on different memory size using R2R-CE (Krantz et al., 2020b) Validation Unseen split.

| | R2R-CE Val Unseen | | | |
|---|---|---|---|---|
| | NE ↓ | OSR ↑ | SR ↑ | SPL ↑ |
| NaVid (Zhang et al., 2024) | 5.47 | 49.0 | 37.0 | 35.0 |
| NaVILA (8 frames) | 5.37 | 57.6 | 49.7 | 45.5 |
| NaVILA (16 frames) | 5.63 | 55.8 | 48.6 | 44.4 |
| NaVILA (32 frames) | 5.74 | 55.9 | 49.5 | 44.1 |
| NaVILA (64 frames) | 5.63 | 60.5 | 50.1 | 45.4 |

Table 10: Ablation study on different memory size using ScanQA dataset (Azuma et al., 2022) Validation split.

| | ScanQA Validation | | | | |
|---|---|---|---|---|---|
| | Bleu-4 ↑ | Rouge ↑ | Cider ↑ | Meteor ↑ | EM ↑ |
| *3D Large Multi-modal Models* | | | | | |
| Scene-LLM (Fu et al., 2024) | 12.0 | 40.0 | 80.0 | 16.6 | 27.2 |
| LEO (Huang et al., 2024b) | 13.2 | 49.2 | 101.4 | 20.0 | 24.5 |
| *2D Vision-Langauge-Action Models* | | | | | |
| NaviLLM (Zheng et al., 2024) | 12.0 | 38.4 | 75.9 | 15.4 | 23.0 |
| NaVILA (8 frames) | 14.8 | 46.4 | 95.1 | 18.7 | 27.0 |
| NaVILA (16 frames) | 15.2 | 48.3 | 99.8 | 19.6 | 27.4 |
| NaVILA (32 frames) | 16.1 | 49.4 | 101.6 | 20.2 | 28.1 |
| NaVILA (64 frames) | **16.9** | **49.3** | **102.7** | **20.1** | **28.6** |

## D  MORE RESULTS ON VLN-CE-ISAAC

Here we show a visualization example highlighting why the Go2 vision policy significantly out-performs the blind policy. As demonstrated in Figure 7, when encountering an obstacle, the VLA, which is not specifically trained for obstacle avoidance, failed to navigate around it effectively. The blind policy, following the VLA's commands without additional sensory input, became stuck at the obstacle. In contrast, the vision-based policy, trained to handle obstacles using LiDAR input, can autonomously avoid dangers even when the high-level VLA model does not detect them.

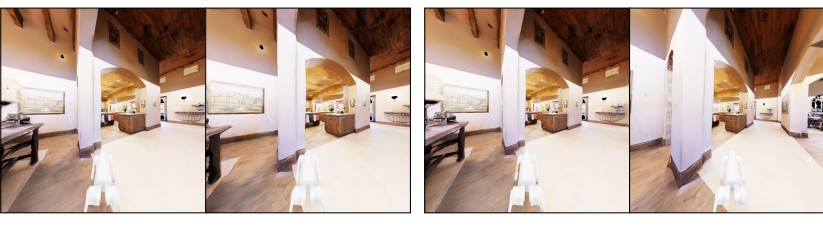

Blind                                    Vision

Figure 7: Comparison between Go2 blind policy and vision policy. The blind policy failed to avoid the obstacles and got stuck. The vision policy detected the obstacle and got around to avoiding it.

## E  IMPLEMENTATION DETAILS FOR VIDEO NAVIGATION TRAJECTORY SUMMARIZATION

We provide the data prompts for our auxiliary task of video navigation trajectory summarization. Following the approach in (Zhang et al., 2024), we construct prompt templates that characterize the LLM as a robot designed for navigation. We process the trajectory videos into history frames, insert the frame tokens into the prompt, and ask the LLM to infer the navigation instructions from the video. This task is designed to enhance the robot's scene understanding and its familiarity with the instruction format.

```
Assume you are a robot designed for navigation.  You are provided
with captured image sequences:  ⟨frame3⟩⟨frame6⟩⟨frame9⟩ Based on
this image sequence, please describe the navigation trajectory of
the robot.
```

### E.1 VLA HYPERPARAMETERS

Please refer to VILA's paper for details on the hyperparameters used in the first two stages. In the instruction fine-tuning stage, we use a learning rate of $1e^{-4}$ with cosine decay and a warm-up ratio of $0.03$. We will release both our training code and data upon paper publication.

## F IMPLEMENTATION DETAILS FOR LOCOMOTION MOTION POLICY

**Reward and randomization:** The reward functions and domain randomization used during Go2 locomotion policy training are listed in Table 11 and Table 12. The robust policy is trained on flat, rough, slope and obstacle terrains shown in Figure 8. LiDAR and height map settings are detailed in Table 13.

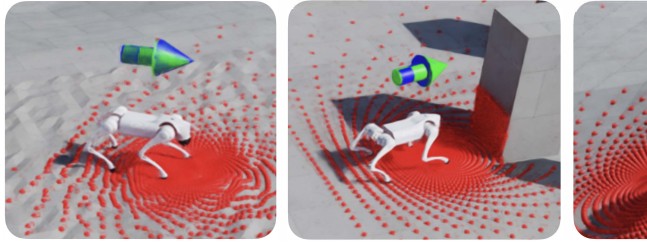

Figure 8: Random rough, obstacle and slope terrain.

Table 11: Reward function parameters for training locomotion policy.

| Reward | Expression | Weight |
|---|---|---|
| Linear velocity tracking | $\exp(-\|v_{xy}^{\text{cmd}} - v_{xy}\|_2^2)$ | 1.5 |
| Angular velocity tracking | $\exp(-(\omega_{\text{yaw}}^{\text{cmd}} - \omega_{\text{yaw}})^2)$ | 1.5 |
| Linear velocity penalty ($z$) | $v_z^2$ | -2.0 |
| Angular velocity penalty ($xy$) | $\|\boldsymbol{\omega}_{xy}\|_2^2$ | -0.05 |
| Flat orientation | $\|\mathbf{g}\|_2^2$ | -2.0 |
| Joint accelerations | $\|\ddot{\boldsymbol{\theta}}\|^2$ | $-2.5 \times 10^{-7}$ |
| Energy | $-\|\tau\dot{q}\|_2^2$ | $-2 \times 10^{-5}$ |
| Body height | $(h^{\text{target}} - h)^2$ | -5.0 |
| Feet slipping | $-\|v_{\text{feet}} \cdot \mathbf{1}[F_{\text{feet}} > 1]\|_2$ | 0.05 |

Table 12: Domain randomization parameters for training locomotion policy.

| Parameter | Value |
|---|---|
| Body Mass | [-3.0, 3.0] |
| Ground Static Friction | [0.4, 4.0] |
| Ground Dynamic Friction | [0.4, 4.0] |
| Motor Strength | [0.9,1.1] |
| System Delay | $[\Delta_t, \Delta_t]$ |

Table 13: LiDAR and Height Map parameters in simulation.

| Parameter | Value |
| --- | --- |
| Channels | 32 |
| Vertical Range (degrees) | (0, 90) |
| Horizontal Range (degrees) | (-180, 180) |
| Horizontal Resolution (degrees) | 4 |
| Voxel Size (m) | 0.06 |
| X Range (m) | [-0.8, 0.2] |
| Y Range (m) | [-0.8, 0.8] |
| Z Range (m) | [0.05, 0.5] |

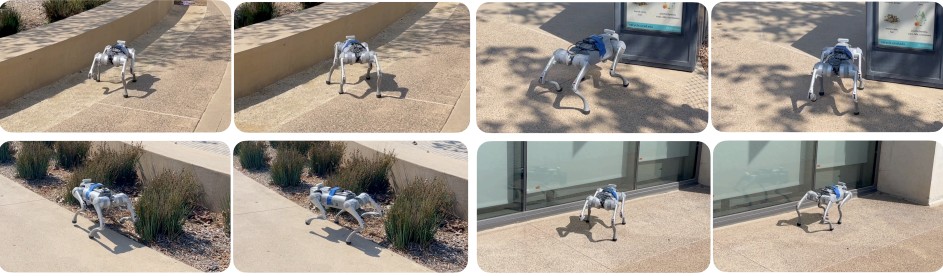

Figure 9: Obstacle avoidance screenshots. Locomotion policy can ensure collision-free in the face of high grass, certain transparent glass, and large objects under strong sunlight. The policy presents robustness on sand and grass terrains.

## G  EXPERIMENTS COMPUTE RESOURCES

**NaVILA Training.**  The first two stages of NaVILA are inherited from VILA (Lin et al., 2024b), which is trained on 16 A100 GPU nodes, with each node having 8 GPUs. The training times for each stage of our 8B model are as follows: connector initialization takes 4 hours, visual language pre-training takes 30 hours. The final visual instruction-tuning stage is experimented on 4 A100 GPU nodes, taking 18 hours.

## H  PARAMETER-EFFICIENT QUANTIZATION

NaVILA's current wait time between each action is about 1 second, which is practical for real-world deployment. All our demonstration videos are shown at 1x speed without any acceleration, accurately reflecting real-world performance. The wait time arises from two factors: image transmission time from Go2 to the server, and the VLA inference time. The transmission time largely depends on the network conditions, while the VLA inference time is approximately 0.6 seconds per sample. We further explore optimization techniques to improve the pipeline efficiency. Specifically, we apply AWQ (Lin et al., 2024a), a state-of-the-art quantization method for VLMs, to the FP16 NaVILA-8B model. By converting it to the W4A16 format (low-bit weight-only quantization), we achieved significant improvements: memory requirements dropped by half, and processing speed improved by about 40%. Most importantly, navigation capabilities remained robust. These optimizations make NaVILA deployable directly on the robot, which will eliminate image transmission time significantly. We leave this as future work.

Table 14: NaVILA quantization results. The computational cost is tested on RTX 4090 with 1737 context tokens and 10 generated tokens, using a sample from R2R-CE as the test case.

| | Computational Cost | | R2R Val-Unseen | | | |
| --- | --- | --- | --- | --- | --- | --- |
| | Total Latency (ms) ↓ | GPU Memory (GB) ↓ | NE ↓ | OS ↑ | SR ↑ | SPL ↑ |
| NaVILA (FP16) | 594.58 | 18.5 | 5.37 | 57.6 | 49.7 | 45.5 |
| NaVILA (W4A16) | **367.80** | **8.6** | 5.66 | 56.8 | 48.2 | 43.6 |

# I LIMITATIONS

While our method shows strong performance, we highlight a failure case in the real world where the robot initially follows the prompt but ultimately fails to reach the correct target position. To further enhance performance, improving generalizability is key, and one potential direction is larger-scale training on more realistic simulations. Additionally, image-based vision-language models require a significant number of tokens, which is computationally intensive and limits the number of history frames processed. This challenge could be addressed with recent advancements in long-context LLMs, allowing for more efficient handling of longer sequences. We leave these as future works.

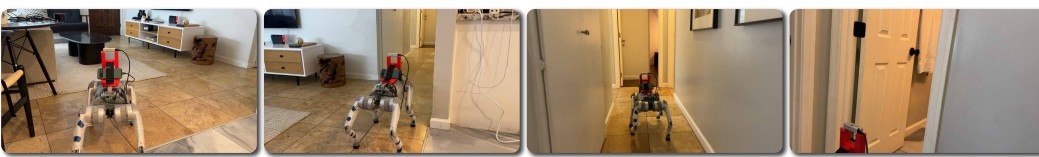

Walk along the hallway and enter the bedroom.

Figure 10: Failure case of NaVILA.

