# OpenReview forum: "NaVILA: Legged Robot Vision-Language-Action Model for Navigation"
_ICLR.cc/2025/Conference — Submitted to ICLR 2025_

### Official Review · Reviewer_mxeA · 2024-10-21

**Soundness:** 4
**Presentation:** 3
**Contribution:** 2
**Rating:** 5
**Confidence:** 4

**Summary:**

This paper proposes NaViLA, a hierarchical framework for vision-language-based legged robot navigation tasks. The framework consists of a high-level vision-language model (VLM) and a vision-based locomotion policy. The high-level VLM converts historical camera observations and user instructions into commands for the low-level locomotion policy to follow. It takes current and previous observations as input and outputs language-based planning instructions. The VLM is fine-tuned from an existing model using an augmented online dataset. The locomotion policy is trained with reinforcement learning (RL) in a physics-based simulation. The proposed method is compared with various benchmarks, and the results show that it outperforms baseline methods. Additionally, a new benchmark for evaluating vision-language navigation (VLN) in a physics-based simulation is introduced. Finally, the proposed method is successfully implemented on a real-world robot.

**Strengths:**

- The proposed method looks reliable and easy to be reproduced.
- Baselines are compared extensively.
- The real world demonstration look good.
- Paper structure is clear and easy to follow.

**Weaknesses:**

- The motivation for using a legged robot is unclear to me. Since the VLN model is trained separately, the low-level control policy should not impact the high-level model, as long as the low-level policy can achieve the desired goal.

- The experimental analysis could be improved. For example, in the 'High-fidelity VLN-CE-Isaac Benchmark' section, the robot with vision outperforms the blind policy. But why is this the case? Are there obstacles that need to be avoided? Should the VLN handle obstacle avoidance? Are there any examples that illustrate the different behaviors of these two policies?

- The overall contribution is limited. From my perspective, the main contribution is fine-tuning a VLM that processes vision input and outputs several predefined intermediate actions.

**Questions:**

Please review the weaknesses section. For the overall contribution, I would appreciate a more detailed summary during the rebuttal.

---

> ### Author Response · Authors · 2024-11-20
> **Response to Reviewer mxeA (1/2)**
>
> We thank the reviewer for their valuable feedback. We address your comments in the following.
>
> ---
>
> **Q:** *The motivation for using a legged robot.*
>
> **A:** Legged robots excel at traversing complex terrains, such as stairs [[video]](https://navila-iclr.github.io/response/videos/general/outdoor_stairs.mp4), uneven surfaces [[video]](https://navila-iclr.github.io/response/videos/general/outdoor_troughs.mp4), and slopes [[video]](https://navila-iclr.github.io/response/videos/general/outdoor_hydrant_turn_left.mp4), where wheeled robots often encounter limitations. To support this motivation, we conduct additional real-world experiments in challenging environments, showcasing the advantages of legged robots in these scenarios. While our high-level planner could be adapted to other robot types, this paper focuses on enabling vision-language navigation specifically for legged robots, addressing the unique challenges and opportunities they present. Please see the General Response for detailed results from the new experiments.
>
> ---
>
> **Q:** *Why does vision outperform the blind policy? Are there obstacles that need to be avoided? Are there any examples that illustrate the different behaviors of these two policies (blind/vision)?*
>
> **A:** Figure 7 in Appendix D ([[video]](https://navila-iclr.github.io/videos/isaac/go2_compare_final.mp4)) illustrates a scenario where a blind policy becomes stuck at a pillar, whereas a vision-based policy successfully navigates around it. This difference is because the high-level VLA model was trained on data generated in the Habitat simulator, which does not account for realistic physics. In Habitat, agents can pass through certain obstacles that would block their path in real-world settings. Therefore, the blind policy, which relies solely on the VLA’s commands without additional sensory input, fails to handle these obstacles. In contrast, the vision-based policy, which incorporates LiDAR input and is trained to account for detailed physical constraints, enables the robot to navigate effectively in such scenarios. This highlights the importance of our proposed VLN-CE-Isaac benchmark, which bridges the gap by addressing these real-world constraints.
>
> ---
>
> **Q:** *Should the VLA handle obstacle avoidance?*
>
> **A:** Yes. The VLA model can handle obstacle avoidance to some extent. It mainly provides directional guidance and avoids large obstacles. However, it cannot manage all obstacles. Smaller or less visible obstacles, which may not be fully captured by the camera or addressed in the VLA's training, fall under the responsibility of the low-level policy. By combining the strengths of the VLA’s high-level guidance with the low-level policy’s obstacle-handling capabilities, NaVILA achieves safer and more efficient navigation in real-world environments.

---

> ### Author Response · Authors · 2024-11-20
> **Response to Reviewer mxeA (2/2)**
>
> **Q:** *The overall contribution is limited. Request detailed summary.*
>
> **A:** Our contributions go beyond fine-tuning a VLM. Below, we outline our detailed contributions and provide supporting evidence from additional experiments.
> 1. Novel Two-Level Framework for Vision-Language Action Models: We introduce a scalable, extensible, and robust framework that integrates high-level VLAs with low-level policy networks. This design addresses key limitations in existing one-stage VLA approaches (e.g. OpenVLA[1]), where **directly fine-tuning a VLM for robot action is not feasible for legged robot navigation**.
>     * Scalable and Extensible: One-stage VLAs require vast robot training data, often obtained through teleoperation or scripted policies, which is expensive and difficult to scale. Our framework separates high-level planning from low-level control to address this, allowing the high-level VLA to leverage diverse data sources (e.g., simulation, real data, human videos). To support this, we conduct additional experiments by adding egocentric human touring videos from YouTube into VLA training, we process the videos into step-wise high-level actions by camera pose estimation. With 2,000 additional videos from YouTube, we observed substantial performance gains on the R2R benchmark, highlighting the scalability and effectiveness of our approach. Please see the General Response for details.
>     * Robustness: One-stage VLAs struggle with low inference frequency (e.g., OpenVLA can only support a control frequency of 5Hz), which cannot satisfy the high-frequency control needs of legged robot navigation (>=50hz). Additionally, managing complex locomotion control is difficult for one-stage VLA models, as most VLAs are designed with a focus on manipulation tasks. In contrast, our high-level VLA model operates at a lower frequency, while a robust low-level controller trained in RL runs at a high frequency, ensuring safe navigation.
> 2. We introduce the first realistic benchmark for the legged robot vision language navigation, VLN-CE-Isaac. VLN-CE-Isaac includes tasks specifically suited for legged robots (e.g., climbing stairs [[video]](https://navila-iclr.github.io/response/videos/djqu/vlnce-isaac-stairs.mp4)). Additionally, VLN-CE-Isaac also simulates realistic physics, as it captures real-world constraints more effectively than prior work, allowing for the evaluation of navigation methods under conditions closer to the real world. To validate this, we conduct an ablation study comparing a blind policy and an Oracle low-level policy (assuming perfect command execution) on VLN-CE-Isaac. Results show a 15% lower success rate on the Go2 setup and a 27% lower success rate on the H1 setup when Oracle policy is not presented. These performance gaps highlight the increased challenges and realism introduced by our benchmark.
> 3. Our results demonstrate state-of-the-art performance on R2R, RxR, and ScanQA. Notably, NaVILA **outperforms all methods** that do not rely on simulator pre-trained waypoint predictors in R2R and RxR, **even when those methods leverage additional inputs** such as depth, panoramic views, and odometry.
> 4. Real robot deployment and strong performance in real-world experiments, validating the effectiveness of NaVILA in challenging environments ([[video]](https://navila-iclr.github.io/response/videos/general/outdoor_hydrant_turn_left.mp4), [[video]](https://navila-iclr.github.io/response/videos/general/indoor_long_horizon.mp4), [[video]](https://navila-iclr.github.io/response/videos/general/indoor_outdoor_transition.mp4)).
>
>
> ---
>
> Please do not hesitate to let us know if you have any additional comments.
>
>
> #### References
> [1] *Kim, M. J., Pertsch, et al., OpenVLA: An Open-Source Vision-Language-Action Model. CoRL, 2024.*

---

> > ### Comment · Reviewer_mxeA · 2024-11-25
> >
> > Thank you for your response.
> >
> > My opinion aligns with Reviewer Lzwz. I don't think the legged locomotion aspect introduces significant novelty. Previous works, such as [1][2], utilized a black-box low-level policy and built the VLM on top of it, while this work employs a pretrained low-level policy. In both cases, the high-level VLM training is independent of the low-level policy. It would be much more compelling to see if the VLM could adapt online to different low-level policies based on the policy's capabilities or the robot platform.
> >
> >
> > References:
> > [1] VLFM: Vision-Language Frontier Maps for Zero-Shot Semantic Navigation, 2023
> > [2] GOAT: GO to Any Thing, 2023

---

> ### Author Response · Authors · 2024-11-23
> **Response to Reviewer mxeA**
>
> Dear Reviewer mxeA,
>
> Thank you once again for the detailed feedback. We are approaching the end of the author-reviewer discussion period. However, there are no responses yet to our rebuttal.
>
> Please feel free to request any additional information or clarification that may be needed. We hope to deliver all the information in time before the deadline.
>
> Thank you!

---

> ### Author Response · Authors · 2024-11-25
> **Response to Reviewer mxeA**
>
> **Latest results:** We will also kindly ask the reviewer to check our latest results here  [[video](https://navila-iclr.github.io/response/videos/acc/indoor_outdoor_transition.mp4),[video](https://navila-iclr.github.io/response/videos/acc/outdoor_troughs.mp4),[video](https://navila-iclr.github.io/response/videos/acc/outdoor_stairs.mp4),[video](https://navila-iclr.github.io/response/videos/acc/outdoor_stone_bench.mp4),[video](https://navila-iclr.github.io/response/videos/acc/outdoor_hydrant_turn_left.mp4),[video](https://navila-iclr.github.io/response/videos/acc/outdoor_hydrant.mp4)], where we show our robot navigating through challenging terrains and complex environments beyond traversing obstacles. Our robot can walk over stones and grass, sidewalks and stairs, and through streets. **This is the first time showing VLN results in unconstrained and diverse indoor/outdoor environments, only enabled by our method.** None of the methods in VLFM [1] or GOAT [2] have provided results like this.
>
> While our VLA model is indeed general-purpose, this generality is essential to handle the complexities unique to legged robots (including Go2 quadruped and H1 humanoid in our experiments). We see legged robot provides a challenging application and it is only enabled with our new VLA model and modular design. Legged robots operate in dynamic and unstructured environments, facing challenges such as uneven terrain, obstacle traversal, and balance maintenance—complexities that are significantly greater than those encountered by wheeled robots. Our method is the first to integrate a general VLA with legged locomotion skills to robustly address these challenges, which we believe constitutes a novel contribution beyond existing works that focus primarily on wheeled robots or drones. **Without our proposed approach, it is unclear how vision-language navigation for legged robots could achieve comparable performance.**
>
> We would like to emphasize while VLFM [1] and GOAT [2] sound similar at first glance, they are fundamentally very different in applications, methods, and model architecture: These works **do not address general vision-language navigation**, instead focusing on object-goal navigation with language descriptions; These methods use CLIP features for inferring explicit affordances and require dynamic construction of environmental maps during deployment. In contrast, NaVILA employs an end-to-end trained Vision-Language Action model for general vision-language navigation, requiring only single-view RGB input and operating without maps, making it more efficient and scalable.

---

> > ### Author Response · Authors · 2024-11-26
> >
> > Dear Reviewer mxeA,
> >
> > We believe our clarifications and additional experiments have fully addressed your concerns. If anything is still unclear, please let us know. Otherwise, we kindly ask you to consider raising your score based on the resolved issues.
> >
> > Thank you!

---

> > > ### Author Response · Authors · 2024-12-01
> > > **Additional Real-world Experiments with Humanoid Robot**
> > >
> > > Dear Reviewer mxeA,
> > >
> > > We have conducted additional real-world experiments using the **humanoid robot G1** to further validate our approach. The attached videos ([#1](https://navila-iclr.github.io/response/videos/g1/g1_indoor_room.mp4), [#2](https://navila-iclr.github.io/response/videos/g1/g1_indoor_trashcan.mp4), [#3](https://navila-iclr.github.io/response/videos/g1/g1_outdoor_statue.mp4), [#4](https://navila-iclr.github.io/response/videos/g1/g1_outdoor_grass.mp4)) demonstrate G1 navigating robustly across diverse indoor and outdoor environments, showcasing its ability to operate effectively in generalized settings.
> > >
> > > Note that these experiments were completed within the short period of time in rebuttal, and importantly, we achieved these results using the same VLA model **without any retraining** for humanoid robot.
> > >
> > > We believe these results strongly demonstrate the significance of our approach and kindly ask you to reconsider the impact of our paper in light of this breakthrough.

---

> > > > ### Author Response · Authors · 2024-12-03
> > > >
> > > > Dear Reviewer mxeA,
> > > >
> > > > We have sent two comments for you to review and have not received any response. Please let us know if any clarification is needed. It is important that you address these comments before the deadline.

---

### Official Review · Reviewer_QouP · 2024-10-28

**Soundness:** 4
**Presentation:** 4
**Contribution:** 3
**Rating:** 8
**Confidence:** 4

**Summary:**

This paper introduces a large vision-language-action model for enabling vision-language navigation for a legged robot. Instead of inferring joint states from image context, authors follow a two step process: (i) predicting the mid-level actions in textual form and (ii) providing these actions to a low-level policy for synthesizing joint action commands. Authors observe the limitations of video-language models (due to lack of large scale data set for the task) and hence take the approach of adapting a VLM to predict actions based on sampled frames from a video. The authors modify prior work VILA to the VLN task by (i) pretraining on generic data sets, (ii) SFT using  VLN data sets and (iii) incorporating auxiliary tasks such as summarization and QA in the associated navigation data. The low-level policy architecture operates on a specific set of skills such as moving forward, turning left etc. and is trained by taking in low-level point cloud maps. Experiments show effectiveness across several benchmarks. Notably performance gain is higher on complex instructions. Finally, the authors also release a new benchmark for the research community.

**Strengths:**

- The paper contributes insights into how to adapt a VLM to predict actions conditioned on present and past visual context. The paper shows how to factor the navigation task as predicting the mid-level actions described in a text form followed by learning a low-level policy. The paper describes the training strategy and SFT data blend and as well as inclusion of auxiliary tasks during training.

- Authors demonstrate their results on common benchmarks, extensively comparing their results to other approaches. Further, a new benchmark is proposed, specifically suited to the problem setting.

- Results indicate higher performance gains on complex instructions. Strong results on legged platforms. Indicative results on the humanoid.

**Weaknesses:**

A key motivation for NaVILA is the ability to generalise using the separation between high-level action prediction and the presence of a low-level policy. The experiments demonstrated are impressive in their sequentiality but do not fully highlight the generalization capacity. Specifically, where the low-level plan changes with high-level inputs and visa versa.

**Questions:**

- The authors emphasize the need for including both the current view and a random sample of past frames (a memory) for predicting actions. In this regard, how much memory (number of samples) is needed for the tasks considered. Presumably, the length would depend on the horizon of the underlying navigation task.

- How does NaVILA respond to action failures. Consider asking a robot to move forward by a distance but the path is either blocked half way or the robot develops a fault. In this case, does the VLA component monitor action execution and replan? More generally, do the low-level actions inform adapation of the high-level plan?

- The core contribution of NaVILA is one of factoring the navigation action generation into predicting mid-level actions in textual form, followed by a low-level join control policy opens doors for behaviour adapation via reasoning. To what extent can the robot perform semantic queries on the world while deciding high-level actions. For example, consider a modification of the action sequence show in top row of figure 1. If one modifies the second phrase to “if you see the stair bar then turn left else go right”, would NaVILA be able to adapt?

---

> ### Author Response · Authors · 2024-11-20
> **Response to Reviewer QouP**
>
> We thank the reviewer for their valuable feedback. We address your comments in the following.
>
> ---
> **Q:** *How much memory (number of samples) is needed for the tasks?*
>
> **A:** We conduct an ablation study to evaluate the impact of memory size (number of history frames) on two tasks: the navigation task using R2R and the spatial understanding task using ScanQA. The results show that for R2R, 8 frames are sufficient to cover most instruction horizons, with limited performance gains from increasing the memory size. In contrast, ScanQA requires a finer-grained memory to recall details such as spatial information of previously seen objects, and performance consistently improves with a larger memory size. **Notably, with a memory size of 64 frames, NaVILA outperforms state-of-the-art 3D-LLM (LEO) across all metrics.** For real-world experiments, we currently use an 8-frame memory size due to latency constraints. While larger memory sizes could potentially improve performance, we leave it as future work.
>
>
> | ScanQA Val | Bleu-4 ↑ | Rouge ↑ | Cider ↑ | Meteor ↑ | EM ↑ |
> |------------|----------|----------|----------|-----------|-------|
> | LEO (3D SOTA) | 13.2 | 49.2 | 101.4 | 20.0 | 24.5 |
> | NaviLLM | 12.0 | 38.4 | 75.9 | 15.4 | 23.0 |
> | NaVILA (8 frames) | 14.8 | 46.4 | 95.1 | 18.7 | 27.0 |
> | NaVILA (16 frames) | 15.2 | 48.3 | 99.8 | 19.6 | 27.4 |
> | NaVILA (32 frames) | 16.1 | 49.4 | 101.6 | 20.2 | 28.1 |
> | NaVILA (64 frames) | **16.9** | **49.3** | **102.7** | **20.1** | **28.6** |
>
>
> | R2R Val Unseen | NE ↓ | OS ↑ | SR ↑ | SPL ↑ |
> |----------------|-------|-------|-------|--------|
> | NaVid | 5.47 | 49.0 | 37.0 | 35.0 |
> | NaVILA (8 frames) | 5.37 | 57.6 | 49.7 | 45.5 |
> | NaVILA (16 frames) | 5.63 | 55.8 | 48.6 | 44.4 |
> | NaVILA (32 frames) | 5.74 | 55.9 | 49.5 | 44.1 |
> | NaVILA (64 frames) | 5.63 | 60.5 | 50.1 | 45.4 |
>
> ---
>
> **Q:** *How does NaVILA respond to low-level action failures? Do the low-level actions inform the adaptation of the high-level plan?*
>
> **A:** No, we currently do not have a mechanism to explicitly inform the high-level planner of low-level action failures. However, this process is implicitly handled within our framework. For example, if the high-level planner commands the robot to move forward a certain distance but the path is blocked midway, the planner will eventually receive history memory frames showing repeated observations of the same scene. These implicit signals can help identify such failures for the high-level planner. It is possible to enhance the system by incorporating these cases into the training set, we leave this refinement for future work.
>
> ---
> **Q:** *Can NaVILA perform high-level semantic queries?*
>
> **A:** Yes, NaVILA is capable of handling complex semantic instructions. As demonstrated in [[video]](https://navila-iclr.github.io/response/videos/QouP/condition.mp4), NaVILA can follow conditional commands such as: "Move forward. If you see a tripod, stop in front of it. Otherwise, turn right and stop in front of the box." These types of instructions are not presented in the training data, yet NaVILA demonstrates strong generalization across vision, language, and action domains by performing them effectively.
>
> ---
>
> Please do not hesitate to let us know if you have any additional comments.

---

> ### Author Response · Authors · 2024-11-23
> **Response to Reviewer QouP**
>
> Dear Reviewer QouP,
>
> Thank you once again for the detailed feedback. We are approaching the end of the author-reviewer discussion period. However, there are no responses yet to our rebuttal.
>
> Please feel free to request any additional information or clarification that may be needed. We hope to deliver all the information in time before the deadline.
>
> Thank you!

---

> > ### Author Response · Authors · 2024-11-25
> > **Official Comment by Authors**
> >
> > Dear Reviewer QouP,
> >
> > **The interactive discussion phase will end in one day (November 26). However, there are no responses yet to our rebuttal.**
> >
> > We hope that our clarifications have addressed your concerns. Please feel free to request any additional information or clarification that may be needed.
> >
> > Thank you!

---

### Official Review · Reviewer_Lzwz · 2024-11-02

**Soundness:** 3
**Presentation:** 3
**Contribution:** 2
**Rating:** 6
**Confidence:** 5

**Summary:**

This paper present NaVILA, a framework that integrates vision-language models with locomotion skills for navigation tasks. It uses high-level language commands and a real-time policy for obstacle avoidance. NaVILA maintains reasoning through language actions, avoiding overfitting and allowing broader task training.

**Strengths:**

1. This paper extends NaVid to legged robot scenarios and demonstrates strong performance in real-world applications.
2. The framework effectively integrates 3D scenes into the policy, enhancing robustness, particularly in real scenarios. This is a commendable attempt.

**Weaknesses:**

1. One issue that remains is the inference time, particularly in the demo where the quadruped robot must wait for the large model to complete inference before executing tasks. This waiting period for model inference can lead to instability in motion, which is a point worth considering for resolution.

2. It is unclear whether the model's performance has been tested in an open environment. Currently, the demos presented are tests conducted in a semi-open environment. The authors could consider introducing environments with greater variance from the dataset to test the model's generalizability in real-world scenarios.

**Questions:**

Please refer to the weakness.

---

> ### Author Response · Authors · 2024-11-20
> **Response to Reviewer Lzwz**
>
> We thank the reviewer for their valuable feedback. We address your comments in the following.
>
> ---
>
> **Q:** *The wait time is too long.*
>
> **A:** The current wait time is about 1 second, which is practical for real-world deployment. The delay arises from two factors: image transmission time from Go2 to the server, and the VLA inference time. The transmission time largely depends on the network conditions, while the VLA inference time is approximately 0.6 seconds per sample. To address this, we explored optimization techniques to improve the pipeline efficiency. Specifically, we applied AWQ, a state-of-the-art quantization method for VLMs, to the FP16 NaVILA-8B model. By converting it to the W4A16 format (low-bit weight-only quantization), we achieved significant improvements: **memory requirements dropped by half**, and **processing speed improved by about 40%**. Most importantly, **navigation capabilities remained robust**. These optimizations make NaVILA deployable directly on the robot, which will eliminate image transmission time significantly. We leave this as future work.
>
>
> | | NE ↓ | OS ↑ | SR ↑ | SPL ↑ |
> |----------------|-------|-------|-------|--------|
> | NaVILA (FP16) | 5.37 | 57.6 | 49.7 | 45.5 |
> | NaVILA (W4A16) | 5.66 | 56.8 | 48.2 | 43.6 |
>
>
> | | Context Stage Latency (ms) ↓ | Context Stage Speed (ms/token) ↓ | Generation Stage Speed (ms/token) ↓ | Average Speed (ms/token) ↓ | Total Latency (ms) ↓ | GPU Memory (GB) ↓ |
> |----------------|------------------------:|---------------------------:|--------------------------------:|------------------------:|-------------------:|------------------:|
> | NaVILA (FP16) | 401.62 | 0.23 | 19.30 | 0.34 | 594.58 | 18.5 |
> | NaVILA (W4A16) | 284.77 | 0.16 | 8.30 | 0.21 | 367.80 | 8.6 |
>
> *(Tested on RTX 4090 with 1737 context tokens and 10 generated tokens, using a sample from R2R as the test case.)*
>
> ---
>
> **Q:** *The waiting period for model inference can lead to instability in motion.*
>
> **A:** During the waiting period for VLA inference, **the robot continues to operate under our low-level RL policy**, with zero velocity as the command input. Our low-level RL policy is specifically trained to handle perturbations, sudden velocity changes, and randomized joint positions, ensuring the robot remains stable even during idle times. This approach is widely adopted in robotics [1, 2, 3] to maintain stable motion in robots while waiting for high-level commands to be processed.
>
> ---
>
> **Q:** *NaVILA’s performance in an open environment. / Generalizability in real-world scenarios.*
>
> **A:** Our submission includes results from laboratory experiments, which are very different from the training data mainly collected from typical indoor houses. In response to the reviewer's concern, **we conduct new real-world experiments in fully open environments**, such as urban streets [[video]](https://navila-iclr.github.io/response/videos/general/outdoor_urban_night.mp4), campus sidewalks [[video]](https://navila-iclr.github.io/response/videos/general/outdoor_hydrant_turn_left.mp4), and courtyards [[video]](https://navila-iclr.github.io/response/videos/general/outdoor_troughs.mp4). These settings add significant variety and challenges, including dynamic objects, different lighting conditions (nighttime), and tough terrains like rocky paths and slopes. Please refer to the General Response for quantitative results.
>
> ---
>
> Please do not hesitate to let us know if you have any additional comments.
>
>
> #### References
> [1] *Lee, J., Hwangbo, J., Wellhausen, L., Koltun, V., & Hutter, M., Learning Quadrupedal Locomotion Over Challenging Terrain. Science Robotics, 2020.*
>
> [2] *Rudin, N., Hoeller, D., Reist, P., & Hutter, M. Learning to Walk In Minutes using Massively Parallel Deep Reinforcement Learning. CoRL, 2022.*
>
> [3] *Miki, T., Lee, J., Hwangbo, J., Wellhausen, L., Koltun, V., & Hutter, M. Learning Robust Perceptive Locomotion for Quadrupedal Robots In the Wild. Science Robotics, 2022.*

---

> > ### Comment · Reviewer_Lzwz · 2024-11-28
> > **Official Comments by Reviewer Lzwz**
> >
> > Thanks for your response.
> >
> > The surprising demo demonstrated the effectiveness of the whole system and addressed my concerns.
> >
> > By the way, I am still quite concerned about the inference issue with this large model because having zero velocity while waiting makes the entire execution process somewhat disjointed. I hope the authors can improve this issue in future work.
> >
> > Overall, I will maintain my positive score.

---

> > > ### Author Response · Authors · 2024-12-02
> > >
> > > Dear Reviewer Lzwz,
> > >
> > > Given the unprofessional behavior of Reviewer djqu, we kindly ask if you can increase the score to support our work.

---

> ### Author Response · Authors · 2024-11-23
> **Response to Reviewer Lzwz**
>
> Dear Reviewer Lzwz,
>
> Thank you once again for the detailed feedback. We are approaching the end of the author-reviewer discussion period. However, there are no responses yet to our rebuttal.
>
> Please feel free to request any additional information or clarification that may be needed. We hope to deliver all the information in time before the deadline.
>
> Thank you!

---

> ### Author Response · Authors · 2024-11-25
> **Official Comment by Authors**
>
> Dear Reviewer Lzwz,
>
> **The interactive discussion phase will end in one day (November 26). However, there are no responses yet to our rebuttal.**
>
> We hope that our clarifications have addressed your concerns and convinced you to consider raising your score. Please feel free to request any additional information or clarification that may be needed.
>
> Thank you!

---

### Official Review · Reviewer_djqu · 2024-11-02

**Soundness:** 2
**Presentation:** 3
**Contribution:** 2
**Rating:** 3
**Confidence:** 4

**Summary:**

This paper presents a method to perform navigation tasks with legged robots. To do so, they train a VLM to translate high-level language instructions into low-level navigation commands, such as "move forward 75 cm" or "turn left 30 degrees," based on current and past single-view RGB image observations. These commands are then executed by a legged robot using a locomotion policy that is trained beforehand in a physics simulator through reinforcement learning to follow velocity commands and avoid obstacles.
The performance of the VLM is evaluated on standard benchmarks for navigation in continuous environments, and a benchmark designed to evaluate navigation and obstacle avoidance is introduced. Finally, the system is deployed on a real Go2 robot.

**Strengths:**

- Real-World Deployment: The authors have successfully implemented their navigation policy on a real legged robot, proving its capability to execute complex navigational tasks in diverse real-world scenarios, including both indoor and outdoor settings.
- Good experimental results in simulated navigation environments: The proposed method outperforms previous approaches for navigation in continuous environments when using only a single RGB camera.

**Weaknesses:**

A significant issue with the paper is that, despite its emphasis on legged locomotion, the navigation skills demonstrated do not inherently require this form of mobility. All the navigation tasks demonstrated could likely be accomplished by wheeled robots without altering the proposed framework. In contrast, prior works on VLA models for legged robots have incorporated capabilities unique to legged robots, such as climbing and crawling under obstacles [1, 2, 3, 4]. The action space of the proposed VLA model is limited to basic navigation commands: “move forward X cm”, “'turn left/right X degrees”, and “'stop”, which abstracts away skills unique to legged robots. The real-world deployment is evaluated in environments where wheeled robots could also navigate effectively. Low-level obstacle avoidance is not exclusive to legged robots. Additionally, it is unclear if the proposed framework for the VLA, which involves supervised fine-tuning from navigation datasets, can handle these kinds of actions only legged robots can perform. The proposed benchmark in simulation, focusing mostly on navigation and collision avoidance, also does not address challenges specific to legged robots.

References:
- [1] Commonsense Reasoning for Legged Robot Adaptation with Vision-Language Models
- [2] Long-horizon Locomotion and Manipulation on a Quadrupedal Robot with Large Language Models
- [3] QUAR-VLA: Vision-Language-Action Model for Quadruped Robots
- [4] Helpful DoggyBot: Open-World Object Fetching usingLegged Robots and Vision-Language Models

**Questions:**

Could the proposed approach be extended to navigation tasks that demand more advanced locomotion skills?

---

> ### Author Response · Authors · 2024-11-20
> **Response to Reviewer djqu**
>
> We thank the reviewer for their valuable feedback. We address your comments in the following.
>
> ---
>
> **Q:** *Can NaVILA handle actions only legged robots can perform (climbing and crawling under obstacles) / Could the proposed approach be extended to navigation tasks that demand more advanced locomotion skills?*
>
> **A:** Yes. For example, as demonstrated in [[video]](https://navila-iclr.github.io/response/videos/general/outdoor_stairs.mp4) [[video]](https://navila-iclr.github.io/response/videos/general/outdoor_stone_bench.mp4), NaVILA successfully climbs up and down stairs, [[video]](https://navila-iclr.github.io/response/videos/general/outdoor_parking_lot.mp4) shows it crossing obstacles, and [[video]](https://navila-iclr.github.io/response/videos/general/indoor_outdoor_transition.mp4) illustrates its ability to transition between indoor and outdoor environments. These tasks highlight its flexibility in performing actions beyond the capabilities of wheeled robots. We conduct new experiments that include these scenarios. Please see the General Response for detailed results.
>
> ---
>
> **Q:** *It is unclear if the proposed framework for the VLA, which involves supervised fine-tuning from navigation datasets, can handle actions only legged robots can perform.*
>
> **A:** NaVILA’s VLA is designed for general navigation, producing simple yet versatile commands like "move forward" or "turn left." Unlike prior works ([1-4]), which typically train a collection of specialized policies as a skill bank to handle specific actions, our approach simplifies execution by relying on a single low-level controller to interpret and act on these commands. This simplicity makes our method more adaptable and easier to deploy across various environments and robot platforms. We have demonstrated that this general policy is sufficient for most everyday navigation tasks, such as **climbing stairs** and **crossing obstacles** (videos in the above response), without relying on expert-designed policies. These results highlight the effectiveness of our method in handling diverse scenarios without the need for task-specific training.
>
> ---
>
> **Q:** *The proposed benchmark in simulation, focusing mostly on navigation and collision avoidance, does not address challenges specific to legged robots.*
>
> **A:** Our benchmark includes two types of legged-specific challenges: First, tasks that wheeled robots typically cannot handle but legged robots excel at (e.g., climbing stairs [[video]](https://navila-iclr.github.io/response/videos/djqu/vlnce-isaac-stairs.mp4)). Second, tasks that are simple for wheeled robots but challenging for legged robots (e.g., careful navigation around furniture [[video]](https://navila-iclr.github.io/response/videos/djqu/vlnce-isaac-furnitures.mp4)). The benchmark's difficulty is evident from our ablation study, where when comparing NaVILAs with a baseline using high-level navigation commands and Oracle’s low-level policy, there remains a gap in performance. This highlights the difficulties in legged navigation. Moreover, as a general benchmark that aimed for **both quadruped and humanoid robots**, we purposely left out special moves like jumping or crawling that work better for certain robot types, instead focusing on common navigation challenges that all legged robots face.
>
> | | NE ↓ | OS ↑ | SR ↑ | SPL ↑ |
> |-------------------|-------|-------|-------|--------|
> | Oracle | 5.25 | 59.8 | 51.3 | 46.9 |
> | NaVILA-Go2-Blind | 6.03 | 49.0 | 36.2 | 33.3 |
> | NaVILA-Go2-Vision | 5.49 | 58.7 | 50.2 | 45.5 |
> | NaVILA-H1-Blind | 7.67 | 33.3 | 24.2 | 21.0 |
> | NaVILA-H1-Vision | 5.86 | 54.6 | 45.3 | 40.3 |
>
> ---
>
> Please do not hesitate to let us know if you have any additional comments.

---

> > ### Author Response · Authors · 2024-11-23
> > **Response to Reviewer djqu**
> >
> > Dear Reviewer djqu,
> >
> > Thank you once again for the detailed feedback. We are approaching the end of the author-reviewer discussion period. However, there are no responses yet to our rebuttal.
> >
> > Please feel free to request any additional information or clarification that may be needed. We hope to deliver all the information in time before the deadline.
> >
> > Thank you!

---

> ### Comment · Reviewer_djqu · 2024-11-24
>
> Thank you for your response.
>
> I maintain my view that this paper does not seem to present a novel contribution regarding the legged robot aspect, despite its emphasis on it.
>
> Deploying a general navigation policy on a legged robot using an intermediate 2D navigation action space is straightforward and not novel (see [5, 6] for recent examples). The proposed VLN method lacks any elements specific to legged robots that would justify framing it as such. For instance, in R2R, many navigation tasks require stair climbing to move between rooms, making a legged robot more suitable than a wheeled robot in some situations, but a quadrotor would also be appropriate. Limiting the set of locomotion skills effectively narrows the scope to general navigation tasks.
>
> This is in contrast to works like [1-4], where tasks typically demand reasoning about the specific capabilities of the legged robot, enabling skills beyond traversing obstacles assumed to be passable in standard navigation.
>
> I think it would be more appropriate to frame the paper as a work centered on VLN, with the deployment on a legged robot serving as experimental validation (similar to [5, 6]) rather than presenting it as a novel contribution.
>
> References:
> - [5] VLFM: Vision-Language Frontier Maps for Zero-Shot Semantic Navigation, 2023
> - [6] GOAT: GO to Any Thing, 2023

---

> ### Author Response · Authors · 2024-11-25
> **Response to Reviewer djqu**
>
> **Results comparing to work like [1-4]:** We respectfully remind the reviewer to see our new results provided in the [rebuttal](https://openreview.net/forum?id=gkDRrvqeWF&noteId=meuPqAs6C5): We have already provided examples of our robot navigating through challenging terrains and complex environments beyond traversing obstacles. Our robot can walk over stones and grass [[video](https://navila-iclr.github.io/response/videos/acc/indoor_outdoor_transition.mp4),[video](https://navila-iclr.github.io/response/videos/acc/outdoor_troughs.mp4)], sidewalks and stairs [[video](https://navila-iclr.github.io/response/videos/acc/outdoor_stairs.mp4),[video](https://navila-iclr.github.io/response/videos/acc/outdoor_stone_bench.mp4)], and through streets [[video](https://navila-iclr.github.io/response/videos/acc/outdoor_hydrant_turn_left.mp4),[video](https://navila-iclr.github.io/response/videos/acc/outdoor_hydrant.mp4)]. In fact, our videos show much more complex scenarios compared to [1-4] if one actually compares the videos: [1] only shows a particular sofa and two different trees; [2] is in a lab setup with QR codes everywhere; [3] is again a lab hallway and room with a few chairs; [4] is a particular sofa and a particular bed in Stanford Robot Center showroom. Our method works anywhere outdoors and in any house we go. This exactly shows the advantages of our VLA it is able to generalize to complex scenes and any place instead of just a few limited scenes. **It shows we have achieved results nobody has achieved before.**
>
>
> While our VLA model is indeed general-purpose, this generality is essential to handle the complexities unique to legged robots. We see legged robot provides a challenging application and it is only enabled with our new VLA model and modular design. Legged robots operate in dynamic and unstructured environments, facing challenges such as uneven terrain, obstacle traversal, and balance maintenance—complexities that are significantly greater than those encountered by wheeled robots. Our method is the first to integrate a general VLA with legged locomotion skills to robustly address these challenges, which we believe constitutes a novel contribution beyond existing works that focus primarily on wheeled robots or drones. **Without our proposed approach, it is unclear how vision-language navigation for legged robots could achieve comparable performance.**
>
>
> We would like to emphasize while VLFM [5] and GOAT [6] sound similar at first glance, they are fundamentally very different in applications, methods, and model architecture: These works **do not address general vision-language navigation**, instead focusing on object-goal navigation with language descriptions; These methods use CLIP features for inferring explicit affordances and require dynamic construction of environmental maps during deployment. In contrast, NaVILA employs an end-to-end trained Vision-Language Action model for general vision-language navigation, requiring only single-view RGB input and operating without maps, making it more efficient and scalable.
>
>
> **Regarding drone or legged robot:** We respectfully point out that quadrotors and legged robots serve fundamentally different purposes. Legged robots can interact with the environment in ways aerial robots cannot, such as potentially manipulating objects on the ground, having a much larger payload for transporting objects, and operating for long hours, navigating confined indoor spaces where flying is impractical or unsafe. Therefore, the existence of quadrotors does not diminish the value of advancing legged robot navigation.

---

> > ### Author Response · Authors · 2024-11-26
> >
> > Dear Reviewer djqu,
> >
> > We believe our clarifications and additional experiments have fully addressed your concerns. If anything is still unclear, please let us know. Otherwise, we kindly ask you to consider raising your score based on the resolved issues.
> >
> > Thank you!

---

> > > ### Author Response · Authors · 2024-12-01
> > > **Additional Real-world Experiments with Humanoid Robot**
> > >
> > > Dear Reviewer djqu,
> > >
> > > We have conducted additional real-world experiments using the **humanoid robot G1** to further validate our approach. The attached videos ([#1](https://navila-iclr.github.io/response/videos/g1/g1_indoor_room.mp4), [#2](https://navila-iclr.github.io/response/videos/g1/g1_indoor_trashcan.mp4), [#3](https://navila-iclr.github.io/response/videos/g1/g1_outdoor_statue.mp4), [#4](https://navila-iclr.github.io/response/videos/g1/g1_outdoor_grass.mp4)) demonstrate G1 navigating robustly across diverse indoor and outdoor environments, showcasing its ability to operate effectively in generalized settings.
> > >
> > > Note that these experiments were completed within the short period of time in rebuttal, and importantly, we achieved these results using the same VLA model **without any retraining** for humanoid robot.
> > >
> > > We believe these results strongly demonstrate the significance of our approach and kindly ask you to reconsider the impact of our paper in light of this breakthrough.

---

> ### Comment · Reviewer_djqu · 2024-12-01
>
> Thank you for your response.
>
> The new experiments on the humanoid robot are a valuable addition.
>
> However, my main concern remains unaddressed. Using an intermediate 2D navigation action space to deploy a VLN model on a legged robot is a common and straightforward approach. In my opinion, presenting this as a novel contribution is problematic, especially as it is a central aspect of the paper.
>
> Prior work on general (robot agnostic) navigation that demonstrates real-world deployment on wheeled or legged robots using similar intermediate 2D navigation action spaces (such as the examples I previously mentioned) could likely be adapted to other platforms.
> The specifics of the VLN task (object-goal navigation, instruction following...) are not directly relevant to my concern.
> Likewise, the ability to overcome certain obstacles is not particularly relevant, as navigation datasets and benchmarks, such as the ones used in this paper, typically assume such obstacles are traversable.
>
> I had some uncertainty about my concern during my initial review, but the rebuttal and discussions have reinforced my confidence in my assessment. I have updated my score to reflect this.

---

> > ### Author Response · Authors · 2024-12-02
> >
> > We sincerely request the reviewer to look at all the replies we wrote to you instead of the partial reply. We want to remind the reviewer that it is YOU, who brought up [1-4] and [5,6]. We have completely addressed the questions on why ours is different and more advanced compared to [1-4] and [5,6]. But the reviewer did not reply to our rebuttal on this, and just repeated again saying this is 2D navigation action space, and this has been done before, **ignoring our explanations on why this is not, and making claims without any evidence**. Even worse, the reviewer just lowered the score **for absolutely no reason**. We hope the reviewer can be more **professional and mature** when writing the comments instead of selectively finding tactical ways to reject the paper by ignoring our answers. This is ICLR's rebuttal. It is serious. We don’t just lower scores or increase scores because “*Oh, I was not sure, I now change my mind.*” This is such a disrespect to the hard work done by the authors. The judgment should be based on facts. The fact is: **We have achieved results no one has shown before.**
> >
> >
> > The reviewer's references to [5,6] clearly misunderstand our work. As we pointed out in the previous response, these papers address object-goal navigation—a far simpler problem than the VLN instruction following. Neither work even attempts to tackle established VLN benchmarks like R2R, making them completely irrelevant as comparisons. Their approaches fundamentally differ from ours: [6] uses deterministic methods such as fast marching algorithms, while [5] uses pre-trained vision models and sensor fusion for waypoint mapping. **Given these significant differences, any attempt to equate our work with [5,6] is baseless and unjustified.** The reviewer's claim about our action space being "common and straightforward" for VLN deployment on legged robots is entirely unsupported and misleading. Please provide the "common work" you mentioned that has tackled the same task using the same approach as ours. **Moreover, reducing our work to just the navigation action space shows a poor understanding of [our contributions](https://openreview.net/forum?id=gkDRrvqeWF&noteId=LnzwQXgGNR).** To suddenly reject our paper solely based on the action space is both unjustifiable and unreasonable. The reviewer's dismissal of obstacle navigation as "not particularly relevant" also directly contradicts their earlier main concern for "lack of advanced locomotion skills." We specifically conducted extensive real-world experiments to address this very concern. Such inconsistency is surprising and suggests a lack of careful consideration of our work.
> >
> > We urge the reviewer to adhere to basic scientific standards: make evidence-based claims, understand the distinctions between methods, and provide consistent, well-founded feedback.

---

### Author Response · Authors · 2024-11-20
**General Response (1/2)**

We thank the reviewers for recognizing the significance of our results (`djqu`, `Lzwz`, `QouP`, `mxeA`), appreciating our efforts toward real-world deployment (`djqu`, `Lzwz`, `mxeA`), and acknowledging the insights provided by our work (`QouP`). Below, we address the reviewers' common feedback, particularly regarding the motivation for using legged robots (`djqu`, `mxeA`) and NaVILA’s generalizability to real-world environments (`Lzwz`).

NaVILA’s strong generalization capabilities are enabled by its modular framework design, which provides exceptional scalability and adaptability to increasingly complex scenes. This design makes NaVILA straightforward to integrate new data sources to enhance performance. For example, we can seamlessly incorporate egocentric human touring videos from YouTube into the VLA training pipeline. These videos were processed into step-wise high-level actions using camera pose estimation, allowing the model to learn effectively from diverse and realistic environments. To validate this, we conduct experiments on the R2R-CE benchmark to measure the performance gains from incorporating YouTube touring videos. As shown in the table below, these experiments demonstrated substantial improvements, with approximately 5% gains in OS, SR, and SPL metrics. This result highlights the scalability and effectiveness of NaVILA’s framework design.

---

| R2R Unseen | NE ↓ | OS ↑ | SR ↑ | SPL ↑ |
|------------|------|-------|-------|--------|
| NaVid | 5.47 | 49.0 | 37.0 | 35.0 |
| NaVILA | 5.37 | 57.6 | 49.7 | 45.5 |
| NaVILA (+ human videos) | **5.22** | **62.5** | **54.0** | **49.0** |

---

Building on this, we conduct more challenging real-world experiments across three environments: Laboratory, House, and Outdoor. These environments included long-horizon tasks traversing narrow passages [[video]](https://navila-iclr.github.io/response/videos/general/indoor_long_horizon.mp4), transitions between indoor and outdoor [[video]](https://navila-iclr.github.io/response/videos/general/indoor_outdoor_transition.mp4), and uneven terrain with small rocks, holes, and troughs in open environments [[video]](https://navila-iclr.github.io/response/videos/general/outdoor_troughs.mp4). In addition to the original experiments, we evaluated 25 instructions, each repeated three times. The quantitative results and accompanying videos below illustrate the complexity of the tasks and the efficacy of our approach. **While NaVILA is compatible with wheeled robots, the motivation for using legged robots lies in their superior ability to handle challenging terrains such as stairs** [[video]](https://navila-iclr.github.io/response/videos/general/outdoor_stairs.mp4), **slopes** [[video]](https://navila-iclr.github.io/response/videos/general/outdoor_hydrant_turn_left.mp4), **and obstacles** [[video]](https://navila-iclr.github.io/response/videos/general/outdoor_parking_lot.mp4)**, where wheeled robots often struggle** [1,2,3]. This advantage makes legged robots a promising platform for advancing vision-language navigation in real-world tasks and has sparked growing industry interest in humanoid and quadruped systems. However, their more complex action space presents greater challenges compared to wheeled robots, making it significantly harder to generalize their behavior to diverse real-world settings. Despite these difficulties, NaVILA achieves strong generalization on legged robots, demonstrating its ability to address these complex scenarios effectively.


| | Lab Simple | Lab Complex | House Simple | House Complex | Outdoor Simple | Outdoor Complex |
|-----------------|------------:|-------------:|---------------:|----------------:|-----------------:|-----------------:|
| GPT-4o | 10/15 | 5/15 | 8/15 | 0/15 | 6/9 | 3/6 |
| NaVILA | 15/15 | 12/15 | 15/15 | 10/15 | 9/9 | 5/6 |

---

> ### Author Response · Authors · 2024-11-20
> **General Response (2/2)**
>
> | Instructions | Challenge | Link |
> |--------------|-----------|------------|
> | *Move forward along the way. Turn left at the yellow fire hydrant. Go forward along the slope and stop in front of the door.* | Complex instruction outdoor, Slope | [[video]](https://navila-iclr.github.io/response/videos/general/outdoor_hydrant_turn_left.mp4) |
> | *Go up the stairs and stop in front of the door.* | Stairs | [[video]](https://navila-iclr.github.io/response/videos/general/outdoor_stairs.mp4) |
> | *Move along the way and stop in front of the stone bench and table.* | Stairs and uneven terrain | [[video]](https://navila-iclr.github.io/response/videos/general/outdoor_stone_bench.mp4) |
> | *Go along the slope and step into the parking lot. Stop in front of the orange cone.* | Rough slope, Cross obstacles | [[video]](https://navila-iclr.github.io/response/videos/general/outdoor_parking_lot.mp4) |
> | *Walk along the pedestrian crosswalk to the other end of the road. Stop in front of the yellow fire hydrant.* | Transition between flat terrain and slope | [[video]](https://navila-iclr.github.io/response/videos/general/outdoor_hydrant.mp4) |
> | *Walk forward along the way. Turn a little left and keep going straight. Stop in front of the red door.* | Uneven terrains with small rocks, holes, and troughs | [[video]](https://navila-iclr.github.io/response/videos/general/outdoor_troughs.mp4) |
> | *Turn right and walk around the stairs until reaching the hallway. Turn a little left, move towards the portrait poster and you'll see an open door. Go straight forward to enter the room until the end of the circular space. Turn left and enter the bathroom. Proceed to the bath mat and step onto it. Turn right and walk forward to reach the weighing machine.* | Long-horizon, Narrow passages | [[video]](https://navila-iclr.github.io/response/videos/general/indoor_long_horizon.mp4) |
> | *Move forward out of the room. Turn right at the end. Proceed to the grass and stop in front of the soccers.* | Indoor-outdoor transition | [[video]](https://navila-iclr.github.io/response/videos/general/indoor_outdoor_transition.mp4) |
> | *Move forward and stop in front of the trash bin.* | Urban nighttime | [[video]](https://navila-iclr.github.io/response/videos/general/outdoor_urban_night.mp4) |
>
>
> #### References
>
> [1] *Lee, J., Hwangbo, J., Wellhausen, L., Koltun, V., & Hutter, M., Learning Quadrupedal Locomotion Over Challenging Terrain. Science Robotics, 2020.*
>
> [2] *Miki, T., Lee, J., Hwangbo, J., Wellhausen, L., Koltun, V., & Hutter, M., Learning Robust Perceptive Locomotion for Quadrupedal Robots In the Wild. Science Robotics, 2022.*
>
> [3] *Agarwal, A., Kumar, A., Malik, J., & Pathak, D., Legged Locomotion in Challenging Terrains using Egocentric Vision. CoRL, 2023.*

---

> ### Author Response · Authors · 2024-11-21
> **Paper Revision Updates**
>
> We have submitted a revised version of our paper, with changes highlighted in red. Below are the updates:
>
> - Added new experiments using real-world data from YouTube human touring videos (L209), which is made possible by our two-level framework design. These experiments highlight the contributions and benefits of our two-stage framework. Notably, we observed a significant performance improvement (Appendix B) with this data, even though we already achieved state-of-the-art performance without it.
> - Added new experiments in real-world scenarios, including outdoor scenes with stairs and complex terrains, to address Reviewer `Lzwz`'s concern about generalizability and Reviewer `djqu`, `mxeA`'s questions on NaVILA's ability to handle tasks suited specifically for legged robots.
> - Added new experiments on quantizing NaVILA into low bits to address reviewer `Lzwz`'s concern regarding inference time (Appendix H).
> - Added new experiments on ablating the effects of different memory sizes (i.e., the number of history frames), as recommended by Reviewer `QouP` (Appendix C).
> - Added new experiments on an Oracle study conducted on our proposed benchmark to address Reviewer `djqu`’s concern (Table 4 and L427).
>
> Again, we sincerely thank the reviewers for their constructive feedback. **We believe that all comments have been addressed in this revision, but are happy to address any further comments from reviewers.**

---

> > ### Author Response · Authors · 2024-12-01
> > **Additional Real-world Experiments with Humanoid Robot**
> >
> > We have conducted additional real-world experiments using the **humanoid robot G1** to further validate our approach. The attached videos ([#1](https://navila-iclr.github.io/response/videos/g1/g1_indoor_room.mp4), [#2](https://navila-iclr.github.io/response/videos/g1/g1_indoor_trashcan.mp4), [#3](https://navila-iclr.github.io/response/videos/g1/g1_outdoor_statue.mp4), [#4](https://navila-iclr.github.io/response/videos/g1/g1_outdoor_grass.mp4)) demonstrate G1 navigating robustly across diverse indoor and outdoor environments, showcasing its ability to operate effectively in generalized settings.
> >
> > Note that these experiments were completed within the short period of time in rebuttal, and importantly, we achieved these results using the same VLA model **without any retraining** for humanoid robot.
> >
> > We believe these results, combined with the experiments presented in the general response above, strongly demonstrate the effectiveness of our approach and highlight the contribution of our paper.

---

> ### Author Response · Authors · 2024-12-03
> **Key Contributions**
>
> We’d like to emphasize our key contributions again and hope the reviewers can recognize our work's significance and potential impact.
>
>
> (1) **Strongly Generalizable VLA Model Trained on Diverse Data.** We tamed a VLM model into a VLA model and trained it on diverse datasets, including question-answering tasks, simulated indoor navigation, and real-world human video data. Our method outperforms prior SOTA by 17% on the R2R and 11.5% on the RxR. NaVILA outperforms all methods that do not rely on simulator pre-trained waypoint predictors, even when those methods leverage additional inputs such as depth, panoramic views, and odometry. Furthermore, we demonstrate for the first time that training directly on human videos improves navigation in continuous environments, while earlier work [1] has only applied them to simpler and discrete settings.
>
>
> (2) **Vision-Based Legged Robot Control Policy.** We proposed the first single-stage vision-based RL policy for legged robots that leverages LiDAR [2]. Previous vision-based approaches often use a two-stage teacher-student framework, which can limit the generalization capabilities of the deployed student policy and complicate the training process. Recent works [3,4,5] show the advantages of one-stage training but are limited to proprioceptive observations. As shown in Figure 4, our LiDAR-based policy ensures safe operation near transparent glass surfaces, whereas other approaches utilizing RGB images and depth cameras often fail. Our policy is tailored specifically for legged robots and excels in navigating rough terrain, slopes, and stairs, outperforming wheeled robots that are limited to flat, structured environments.
>
>
> (3) **Comprehensive Evaluation Benchmark.** We introduced VLN-CE-Isaac, a benchmark providing higher dynamic fidelity compared to current alternatives. As highlighted in Section 3.2, existing benchmarks based on simulators like Habitat lack the precision needed for low-level robotic control. Deploying pipelines directly in the real world is not only costly—considering that humanoid robots can exceed $100k—but also risky, as unrefined policies can lead to hardware damage. Moreover, testing navigation across diverse scenarios for various robot types is highly time-intensive. VLN-CE-Isaac addresses these limitations by providing a safe, efficient, and scalable testing platform, laying a critical foundation for advancing robotic navigation research across various robot types and scenarios.
>
>
> (4) **Real-World Deployment on Legged Robots with Strong Performance.** We validated the effectiveness of NaVILA in challenging real-world environments using both quadruped and humanoid robots. Our results demonstrated robust performance and exceptional generalization across diverse settings, leveraging VLA's generalization strength and the low-level policy's robustness. The results we achieved represent a significant milestone, showcasing capabilities that have never been demonstrated before.
>
> #### References
>
> *[1] Learning Vision-and-Language Navigation from YouTube Videos*
>
> *[2] https://www.unitree.com/LiDAR*
>
> *[3] DreamWaQ: Learning Robust Quadrupedal Locomotion With Implicit Terrain Imagination via Deep Reinforcement Learning*
>
> *[4] Hybrid Internal Model: Learning Agile Legged Locomotion with Simulated Robot Response*
>
> *[5] Learning H-Infinity Locomotion Control*

---

> > ### Comment · Reviewer_djqu · 2024-12-03
> >
> > I would like to quickly react to this claim by the authors in the previous message:
> >
> > "(2) Vision-Based Legged Robot Control Policy. We proposed the first single-stage vision-based RL policy for legged robots that leverages LiDAR"
> >
> > The 2021 paper "Learning to Walk in Minutes Using Massively Parallel Deep Reinforcement Learning" by Rudin et al.  for example also proposed a single-stage reinforcement learning policy where the robot perceives the surrounding terrain using a heightmap from LiDAR sensors. The approach described in Section 2.2 appears very similar to Rudin et al.
> >
> > I didn’t highlight this in my review to focus on what I believe is a more central and critical aspect of the paper.

---

> > > ### Author Response · Authors · 2024-12-04
> > >
> > > No, this is not true. The 2021 paper is **NOT** a truly vision-based policy. During training, it does **NOT** use LiDAR sensors but instead queries pre-defined terrain heights—heuristic information hard-coded into the simulator—rather than interacting directly with the environment. This reliance on static data leads to a significant sim-to-real gap, as the authors themselves admit as a limitation:
> > >
> > > >  “Unfortunately, this height map *(real)* is far from perfect, which results in a decrease in robustness between simulation and reality.”
> > >
> > > In contrast, our work represents a fundamentally different approach by directly interacting with the environment through IsaacSim’s LiDAR sensors. Through raycasting, we dynamically construct height maps in each step based on actual terrain feedback, enabling a truly **vision-based approach**. Our method has demonstrated robust performance in the real world, as evidenced by our real-world experiment videos.
> > >
> > > The reviewer’s repeated claim that “Paper A appears very similar to your work” once again lacks any specifics or evidence to justify it. We respectfully ask the reviewer not to mischaracterize our work and others.

---

### Meta-Review · Area_Chair_famD · 2024-12-22

**Metareview:**

The paper proposes NaVILA, a vision-and-language navigation (VLN) framework that integrates a vision-language-action (VLA) model with locomotion skills for legged robot locomotion. Rather than mapping natural language instructions to low-level actions, NaVILA adopts a two-stage architecture in which the vision-language model reasons over the instructions and the robot's observations to infer primitive (mid-level) text-based actions that then serve as input to a low-level policy that generates joint action commands. For the high-level VLM, NaVILA fine-tunes VILA, an existing image-based VLM, based on available navigation datasets. The low-level locomotion policy is pre-trained in simulation via reinforcement learning. The paper evaluates NaVILA on a series of simulation-based benchmarks and demonstrates its performance on a variety of challenging, real-world legged locomotion tasks.

The paper was reviewed by four referees and received a healthy amount of attention during the rebuttal and the reviewer-AC discussion phases. The paper is highly topical---the use of vision-language models for robot reasoning continues to receive a lot of attention in the robot learning community. All four reviewers agree that a key strength of the paper is NaVILA's successful deployment on real robots, with demonstrations of robots locomoting in a number of challenging scenarios. The authors expanded on these demonstrations with results on humanoid robots during the rebuttal period, which the AC and reviewers appreciated as being no small feat. At least three of the reviewers further emphasized NaVILA's strong performance relative to contemporary baselines on simulation-based benchmarks. Others appreciated the means by which the low-level policy incorporates 3D scene information as well as their perception that the method can be easily deployed in new scenarios.

However, several reviewers raised concerns about the significance of the paper's contributions relative to the current state-of-the-art and the motivation for using legged robots. Some reviewers felt that the paper over-states the novelty of the bi-level architecture and the application to legged robots. There was concern that the contributions relative to NaVid are relatively incremental and while NaVid was applied to wheeled robots, it is not clear that it could not readily be extended to legged robots using a similar low-level policy given the similar nature of the mid-level action space. Additionally, the reviewers questioned the contributions over other works that similarly employ a bi-level architecture that builds a vision-language model on top of low-level policies. That said, it is clear to the AC that the real robot scenarios considered here are more complex than those demonstrated in other papers, however what was not made sufficiently clear is what unique aspects of NaVILA make this possible. In their rebuttal, the authors emphasize NaVILA's modularity as being key to its generalizability, however it is not obvious why a framework like NaVid can not be considered to be similarly modular. The AC acknowledges that that NaVILA application to legged robots (compared with NaVid) and its demonstrations on more challenging locomotion scenarios (compared with existing two-stage legged robot methods), but would like to see a clearer discussion of the significance of this contribution. Related, several reviewers felt that while the paper developes NaVILA in the context of legged robots, it is not apparent in what ways the framework is specific to legged locomotion as opposed to using legged robots as platform for experimental evaluation. Indeed, the authors acknowledge the general-purpose nature of NaVILA and claim that this generality is critical in light of the challenges of legged locomotion, however the reasons for this are not sufficiently clear. Less critical, there was a discussion with the reviewers about the claim that the paper proposes the first vision-based RL policy that employs LiDAR in light of the work of Hoeller et al. [1].

Overall, the AC believes that if the paper were clearer in how it is positioned in the context of contemporary methods that adapt vision-language models for (legged) robotics---whether it is with regard to how NaVILA is specific to legged locomotion or that legged robots serve as the platform for experimental evaluation---it would provide a valuable contribution to the community.


[1] D. Hoeller, N. Rudin, D. Sako, and M. Hutter, "ANYmal Parkour: Learning Agile Navigation for Quadrupedal Robots", arXiv preprint arXiv:2306.14874

**Additional Comments On Reviewer Discussion:**

The AC acknowledges the authors' concerns regarding the responses of some of the reviewers during the author-reviewer discussion period. The AC recognizes that it is often easier for reviewers to focus on a paper's weaknesses rather than its strengths and as someone who has been in their shoes, the AC understands the frustration of feeling that a reviewer disregarded key aspects of their rebuttal.

The AC discussed the paper with all four reviewers after the author-reviewer discussion period. The meta-review and recommendation were made after considering this discussion, the authors' responses to the initial reviews, and feedback provided to the AC.

---

### Decision · Program_Chairs · 2025-01-22

Reject